# Weekend and weekday associations between the residential built environment and physical activity: Findings from the ENABLE London study

**Christelle Clary[1], Daniel Lewis[1], Elizabeth S. Limb[2], Claire M. Nightingale[2], Bina Ram[2], Alicja R. Rudnicka[2], Duncan Procter[3,4], Angie S. Page[3,4], Ashley R. Cooper[3,4], Anne Ellaway[5], Billie Giles-Corti[6], Peter H. Whincup[2], Derek G. Cook[2], Christopher G. Owen[2]\*, Steven Cummins[1]**

**1** Department of Public Health, Environments and Society, London School of Hygiene and Tropical Medicine, London, United Kingdom, **2** Population Health Research Institute, St George's, University of London, London, United Kingdom, **3** Centre for Exercise, Nutrition and Health Sciences, School of Policy Studies, University of Bristol, Bristol, United Kingdom, **4** National Institute for Health Research Bristol Biomedical Research Centre, University Hospitals Bristol NHS Foundation Trust and University of Bristol, Bristol, United Kingdom, **5** MRC/ SCO Social and Public Health Sciences Unit, University of Glasgow, Glasgow, United Kingdom, **6** NHMRC Centre of Research Excellence in Healthy Liveable Communities, RMIT University, Melbourne, Victoria, Australia

\* cowen@sgul.ac.uk

**Data Availability Statement:** Data are available from https://doi.org/10.24376/rd.sgul.12436274 There are restrictions on the availability of these

## Abstract

### Background

We assessed whether the residential built environment was associated with physical activity (PA) differently on weekdays and weekends, and contributed to socio-economic differences in PA.

### Methods

Measures of PA and walkability, park proximity and public transport accessibility were derived for baseline participants (n = 1,064) of the Examining Neighbourhood Activities in Built Living Environments in London (ENABLE London) Study. Multilevel-linear-regressions examined associations between weekend and weekday steps and Moderate to Vigorous PA (MVPA), residential built environment factors, and housing tenure status as a proxy for socio-economic position.

### Results

A one-unit decrease in walkability was associated with 135 (95% CI [28; 242]) fewer steps and 1.2 (95% CI [0.3; 2.1]) fewer minutes of MVPA on weekend days, compared with little difference in steps and minutes of MVPA observed on weekdays. A 1km-increase in distance to the nearest local park was associated with 597 (95% CI [161; 1032]) more steps and 4.7 (95% CI [1.2; 8.2]) more minutes of MVPA on weekend days; 84 fewer steps (95% CI [-253;420]) and 0.3 fewer minutes of MVPA (95%CI [-2.3, 3.0]) on weekdays. Lower

data due to the signed consent agreements around data security, which only allow access to external researchers for research monitoring purposes. Requestors wishing to access the data for the purposes of replicating or checking our analyses should contact the SGUL RDM service at researchdata@sgul.ac.uk.

**Funding:** This research is being supported by project grants from the UK National Prevention Research Initiative (MR/J000345/1) and the UK National Institute for Health Research (NIHR; 12/211/69). Diabetes and obesity prevention research at St George's, University of London is supported by the NIHR Collaboration for Leadership in Applied Health Research and Care, South London. CMN is supported by the Wellcome Trust Institutional Strategic Support Fund (204809/Z/16/Z). BR was supported by a PhD studentship from St George's, University of London. ARC and ASP are supported by the NIHR Biomedical Research Centre at University Hospitals Bristol National Health Service Foundation Trust and the University of Bristol. AE is funded by the Medical Research Council as part of the Neighbourhoods and Communities Programme (MC_UU_12017–10). BG-C is supported by a National Health and Medical Research Council Principal Research Fellowship (1107672). The funders had no role in study design, data collection and analysis, decision to publish, or preparation of the manuscript.

**Competing interests:** No authors have competing interests.

public transport accessibility was associated with increased steps on a weekday (767 steps, 95%CI [–13,1546]) compared with fewer steps on weekend days (608 fewer steps, 95% CI [–44, 1658]). None of the associations between built environment factors and PA on either weekend or weekdays were modified by socio-economic status. However, socio-economic differences in PA related moderately to socio-economic disparities in PA-promoting features of the residential neighbourhood.

## Conclusions

The residential built environment is associated with PA differently at weekends and on weekdays, and contributes moderately to socio-economic differences in PA.

## Introduction

Physical activity (PA) is a protective factor for a wide range of physical and psychological disorders [1–3]. Current population levels of PA are too low in the UK, with 37% of adults aged 16 years or more not meeting recommended levels of activity of at least 150 minutes of moderate-intensity per week [4]. Individuals from more disadvantaged socio-economic groups are less physically active [4], both in terms of daily steps [5] and moderate to vigorous physical activity (MVPA) [6, 7]. Recently, epidemiological research has increasingly incorporated socio-ecological models that acknowledge the role of the built environment, especially the local residential environment [8], in influencing PA behaviours [9]. The extent to which the residential built environment contributes to individual socio-economic differences in PA has been explored by drawing on two hypothetical pathways: (i) a deprivation-amplification effect [10], whereby disadvantaged individuals are less exposed to health-promoting facilities in their residential neighbourhood, and (ii) a moderating effect, by which socio-economic groups use the physical activity facilities available in their neighbourhood differently [11]. Regarding the deprivation-amplification pathway, some UK studies have reported that the most affluent urban areas have the poorest accessibility to recreational PA facilities [12–15], but others the most deprived [16]. As for the moderating effect pathway, some studies have reported that the association between the residential built environment and PA was moderated by the socio-economic position [17, 18], whereas others did not find such evidence [19, 20]. These mixed findings likely reflect diverse local realities, that may owe to local population specificities or regional policy interventions. More research is needed to depict a comprehensive, overarching view of how built environment, socio-economic status and PA behaviours interrelate, and, in turn, deliver effective 'contextually sensitive' policy interventions [21].

Residential neighbourhood characteristics positively associated with PA include increased walkability [22], better access to greenspace [23, 24] and public transport [23]. However, these UK findings rely on a limited number of studies where the evidence is mixed. For instance, a study from Combes et al. [24] shows a decrease in odds of achieving PA recommendations associated with increasing distance to green space from home adult (16+ years) residents in Bristol (UK). Yet, Foster et al. found a null association between proximity to the nearest public open/green space and PA levels in adults aged 45–74 years from the European Investigation into Cancer study (EPIC) Norfolk study [25]. More work is required to establish whether specific features of the residential built environment shape PA behaviours in the UK, and whether these features might help explain socio-economic differences in PA.

Associations between residential PA facilities and PA have been shown to vary by time of the day and day of the week [26]. Daily variations in number of steps and minutes of MVPA have been reported in English adults, with more PA on weekdays compared to weekends [5, 27]. UK time use surveys highlight how individuals spend more time travelling and in paid work or studying on weekdays, and more time in leisure activities and domestic work on weekends [28, 29]. Use of the residential built environment may therefore be expected to be different on weekdays and weekends. This suggests we should also consider weekday and weekend variations in PA when seeking to better understand the contribution of the residential built environment to PA and the role of socio-economic status within it.

### Aims

The aim of this London (UK) based study is twofold. First, to assess whether the residential built environment is associated with the number of daily steps, and the amount of daily MVPA (min) accumulated, on weekdays and weekend days. Second, to explore two pathways in which the built environment may contribute to household-level socio-economic differences in PA levels on weekdays and weekend days: (i) a moderating effect, whereby different socio-economic groups relate differently to the physical activity facilities available in their neighbourhood; (ii) a deprivation-amplification effect [10], by which reduced access to PA-promoting facilities for disadvantaged individuals translate into lower levels of PA.

## Methods

### Study design and participants

The Examining Neighbourhood Activities in Built Living Environments in London (ENABLE London) study is a controlled before and after natural experiment aimed at establishing whether PA behaviours improve among individuals relocating to East Village [30]. East Village, formerly the London 2012 Olympics Athletes' Village, is a planned mixed-use residential neighbourhood development designed to both encourage active living and promote social mixing through three types of housing tenure: "social", "intermediate" (including affordable rent, shared ownership and shared equity), and "market-rent". In this paper we use baseline measurements of PA and sociodemographic data from the ENABLE London study on adults prior to any relocation into East Village.

Participants were recruited from those seeking "social", "intermediate" and "market-rent" accommodation in East Village. The sample consisted of 1,278 individuals aged 16 years and over clustered in 1,006 households, with response rates of 52%, 57% and 58% for social, intermediate and market-rent housing seekers, respectively. Baseline assessments were carried out at the participant's place of residence between January 2013 and December 2015, prior to any potential move to East Village. Participants were asked to complete a questionnaire and to wear a hip-mounted accelerometer (ActiGraph GT3X+) for one week, that they were to remove only when in water or sleeping. Full ethical approval was obtained from the City Road and Hampstead Ethical Review Board (REC Reference 12/LO/1031). All participants provided written informed consent.

Of the 1,278 study participants, we excluded those who lived outside Greater London and without sufficient accelerometer data (see flow diagram of participation in Fig 1). The 214 individuals who were excluded from the analytical sample had similar characteristics to the 1,064 who were included, in terms of housing status (Pearson Chi-square, p = 0.172), age (Pearson Chi-square, p = 0.273), sex (Pearson Chi-square, p = 0.06) and ethnicity (Pearson Chi-square, p = 0.181) (S1 Table). Of the 1,064 participants clustered in 842 households retained for analyses, 442 were seeking relocation into a social, 436 into intermediate, and 186

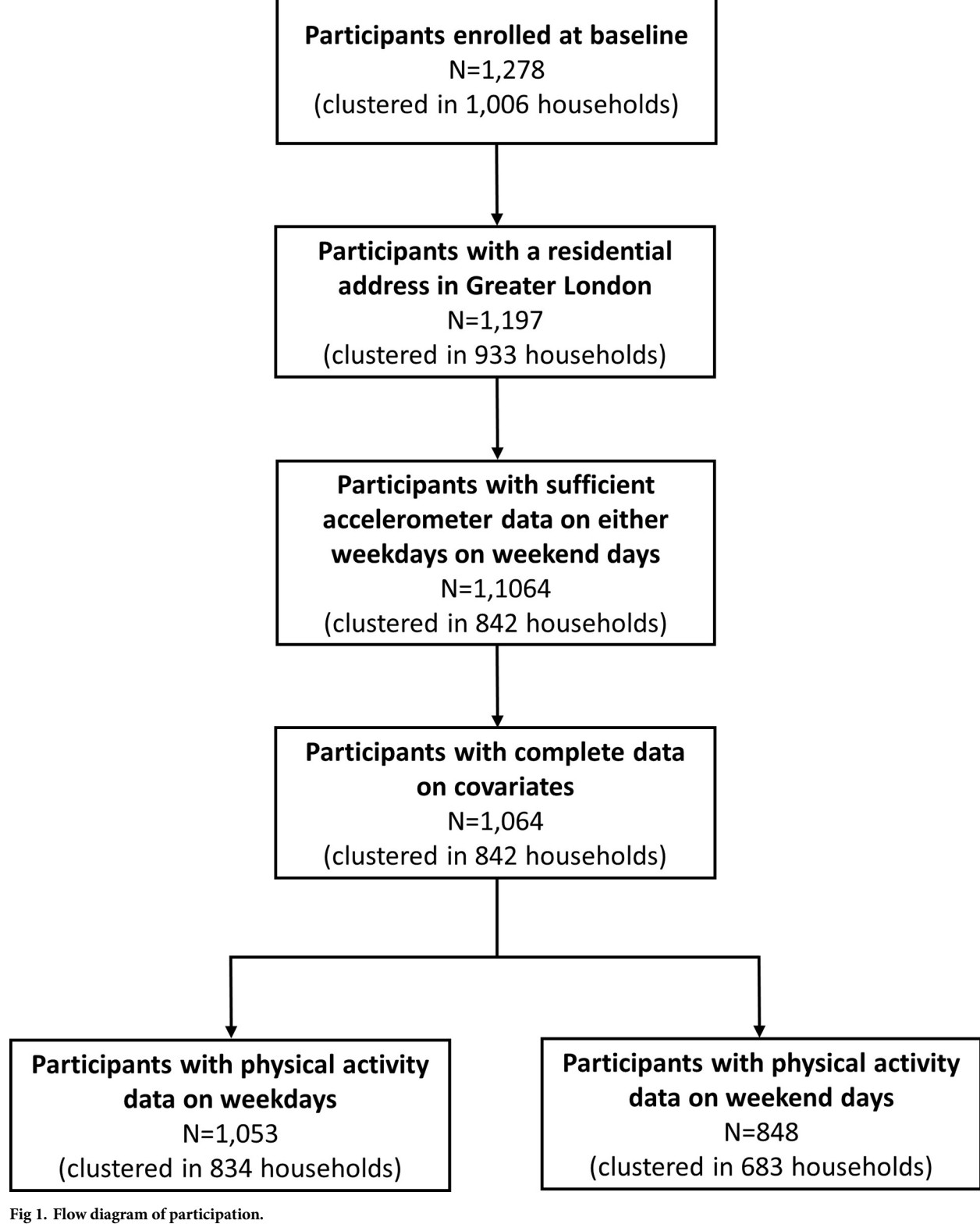

**Fig 1. Flow diagram of participation.**

into market-rent accommodation; 1,053 participants had PA data on weekdays, 848 on weekends, and 837 on *both* weekdays *and* weekends.

## Variables

**Accelerometer-derived PA outcomes.** Participants were asked to wear a hip-mounted ActiGraph GT3X+ accelerometer for 7 consecutive days during waking hours. This provided daily measures of steps and time spent in moderate-to-vigorous physical activity (MVPA), based on the standard threshold of $\geq 1952$ counts per minute [31]. We excluded days of accelerometer data where the registered wear time was less than 540 minutes. Multilevel linear regression models were fitted to allow for repeated measurements of PA, level 1 was day within individual and level 2 was individual. Daily steps (minutes of MVPA) was regressed on day-order-of-wear, day-of-week and month-of-wear, and an unbiased estimate of average daily steps (minutes of MVPA) obtained for each participant for weekdays and weekend days separately [5].

**Environmental variables.** Participants were geocoded to the centroid of the footprint of their building of residence, using Ordnance Survey (OS) AddressBase Premium (March 2015) to match participants' declared residential addresses with XY coordinates. Residential locations were then used to derive a range of built environment factors hypothesised to be associated with PA, as follows.

*Residential neighbourhood walkability*. Walkability scores were calculated by summing the z-scores of three GIS-derived variables aggregated within a 1km-street network home-centred buffer: street connectivity, land use mix and net residential density. *Street connectivity* was defined as the number of 3 or more branch road junctions per street-kilometre. *Land use mix* was derived as the evenness of distribution of square footage of residential, commercial, office, entertainment and institutional building footprints, based on existing literature [22, 32]. *Net residential density* was defined as the unique residential addresses per squared kilometre of building footprint devoted to residential use (i.e. residential building and attached gardens). Land use mix and net residential density variables were log-transformed before deriving the z-scores due to their skewed distributions. The choice of 1km-street-network-home-centred buffers was motivated by two considerations. First, destinations that are within 1km from home have been defined as reachable by foot in the literature [33]. In the absence of more personalised measures of the residential neighbourhood, 1km was thus judged as a good aggregation unit to encompass the opportunities available in the residential area of each participant. Second, a 1km buffer has been used in many studies that found significant associations between features of the built environment and PA outcomes (e.g. [34, 35]). Additionally, a 1-km street network is equivalent to a 10–15 minute walk from home/centre of the buffer to its boundaries.

*Proximity from home to parks*. Using data from Greenspace Information for Greater London (GiGL) [36], proximity to three types of park – "metropolitan", "district", and "local" – was calculated on the basis of street-network distance from the home address to the nearest entrance for each park type. Park type is derived from the Greater London Authority (GLA) London Plan March 2016 [37] and is based on park size and the number and type of facilities they provide. "Metropolitan" parks are the largest and have the most amenities, and "local" parks are the smallest and least well-equipped of these three types (S2 Table). Where there were missing entrance points to parks in the GiGL database (n = 22, i.e. 2.9%), they were manually geocoded based on visual data drawn from Google Maps.

*Public transport accessibility*. Each ENABLE London participant was assigned a PTAL (Public Transport Accessibility Level) score based on the closest location to their place of residence

where a PTAL value was made available by Transport for London (TfL) [38]. PTAL is an averaged measure of the densities of, and the frequency and reliability of service at, the London public transport access points (trains, buses, underground, Docklands Light Railway (DLR), trams) [39], that is being commonly used in London [40]. It is classified into six-value ranges (lower scores reflecting poorer accessibility), whose scores are available for the centroid of each 100m by 100m cell of a grid covering the whole of Greater London [39]. We collapsed the six-value ranges into three categories (Low: PTAL scores 0, 1a, 1b; Intermediate: PTAL scores 2,3,4; High: PTAL scores 5, 6a,6b) in order to increase the number of participants per category, especially for those residing in areas with the lowest accessibility scores.

Data sources and versions used for computing these environmental variables are detailed in S3 Table.

**Covariates and moderators.**    Covariates included sex (female, male), age group ([16-24y], [25-34y], [35-49y], [50y+]), ethnicity (White, Black, Asian, Mixed/Other), and the type of housing tenure participating households sought to relocate to in East Village ("social", "intermediate", and "market-rent"). Aspirational housing tenure was used as a marker of household socio-economic status, with "social" and "market-rent" referring to the most deprived and affluent groups, respectively. This was for two reasons. First, housing tenures are allocated largely based upon individual and household social and economic characteristics (S4 Table). Housing tenure has therefore commonly been used as a proxy for socio-economic status in the wider epidemiological literature [41]. In our sample, 67% of those seeking social housing were currently living in social housing accommodation, with the remainder being mostly on social housing waiting lists. Of those seeking intermediate or market-rent accommodation, 64% were currently living in privately rented accommodation, with the remainder largely living with relatives or friends. Considering aspirational housing tenure as a socio-economic marker is supported by the observation that ENABLE London participants seeking relocation into social housing are also more likely to be unemployed, less educated and from ethnic minorities (i.e. classic markers of socio-economic vulnerability) compared to those seeking relocation into affordable and market-rent housing (Table 1). Second, literature has highlighted associations of both current housing tenure and aspirational housing tenure with a variety of health outcomes [5, 30, 42, 43] (e.g. BMI, mental health, morbidity and mortality), including PA [44]. Seekers of social housing were used as the reference group, because it enrolled a larger number of participants.

Because household income, occupational status, presence of children in the household and education level were part of the eligibility criteria for housing in East Village (S4 Table) and/or highly correlated to aspirational housing status, these variables were not adjusted for in the final regression models to avoid over-adjustment. Other hypothesised covariates, including whether a car was available for use in the household and attitude towards PA, were excluded after initial consideration, either because they were not found to be associated with exposures and outcomes in bivariate analyses or because their addition to fully adjusted models did not alter estimated coefficients appreciably.

Aspirational housing tenure was also tested as a moderating factor of the associations between residential built environment factors and PA outcomes.

## Statistical analyses

Multi-level linear regression models were used to examine the association between daily PA (steps and minutes of MVPA) and residential built environment variables (walkability, distance to parks and accessibility to public transport) on weekdays and weekend days separately. Average daily steps (minutes of MVPA) were regressed on each built environment variable

**Table 1. Baseline demographic, residential built environment and daily physical activity by aspirational housing tenure in the London-ENABLE study.**

| | Total | | Social | | Intermediate | | Market rent | | |
|---|---|---|---|---|---|---|---|---|---|
| N | 1064 | | 442 | | 436 | | 186 | | |
| | n | (%) | n | (%) | n | (%) | n | (%) | p-value |
| **Sex:** female | 621 | (58%) | 330 | (75%) | 210 | (48%) | 81 | (44%) | <0.001 |
| **Age** | | | | | | | | | <0.001 |
| 16–24 | 222 | (21%) | 95 | (21%) | 77 | (18%) | 50 | (27%) | |
| 25–34 | 464 | (44%) | 112 | (25%) | 254 | (58%) | 98 | (53%) | |
| 35–49 | 310 | (29%) | 198 | (45%) | 92 | (21%) | 20 | (11%) | |
| 50+ | 68 | (6%) | 37 | (8%) | 13 | (3%) | 18 | (10%) | |
| **Ethnic group** | | | | | | | | | <0.001 |
| White | 511 | (48%) | 83 | (19%) | 301 | (69%) | 127 | (68%) | |
| Black | 270 | (25%) | 210 | (48%) | 46 | (11%) | 14 | (8%) | |
| Asian | 172 | (16%) | 90 | (20%) | 61 | (14%) | 21 | (11%) | |
| Mixed / Other | 111 | (10%) | 59 | (13%) | 28 | (6%) | 24 | (13%) | |
| **Qualifications** | | | | | | | | | <0.001 |
| Degree or equivalent | 612 | (58%) | 101 | (23%) | 364 | (84%) | 147 | (79%) | |
| Intermediate qualification | 318 | (30%) | 243 | (55%) | 51 | (12%) | 24 | (13%) | |
| Other/None | 132 | (12%) | 97 | (22%) | 20 | (5%) | 15 | (8%) | |
| **NS-SEC [1]** | | | | | | | | | <0.001 |
| Higher managerial/professional | 488 | (46%) | 50 | (11%) | 309 | (71%) | 129 | (69%) | |
| Intermediate occupations | 158 | (15%) | 54 | (12%) | 70 | (16%) | 34 | (18%) | |
| Routine/manual occupations | 139 | (13%) | 107 | (25%) | 24 | (6%) | 8 | (4%) | |
| Unemployed/Economically inactive | 270 | (26%) | 225 | (52%) | 30 | (7%) | 15 | (8%) | |
| **Residential Built Environment Factors** | | | | | | | | | |
| Walkability, mean (95% CI) | 0.04 | (-0.12, 0.20) | -0.51 | (-0.70, -0.31) | 0.21 | (-0.05, 0.46) | 0.94 | (0.48, 1.39) | <0.001 |
| Distance to metropolitan parks (km), median (IQR) | 2.16 | (1.21, 3.48) | 2.53 | (1.43, 3.71) | 1.90 | (0.95, 2.99) | 1.85 | (1.04, 3.15) | <0.001 |
| Distance to district parks (km), median (IQR) | 2.21 | (1.35, 3.08) | 2.39 | (1.72, 3.32) | 2.04 | (1.07, 2.88) | 2.00 | (1.39, 2.75) | <0.001 |
| Distance to local parks (km), median (IQR) | 0.75 | (0.44, 1.18) | 0.62 | (0.38, 0.98) | 0.88 | (0.48, 1.35) | 0.86 | (0.48, 1.37) | <0.001 |
| Public transport accessibility, n (%) | | | | | | | | | <0.001 |
| Low | 96 | (9%) | 47 | (11%) | 37 | (8%) | 12 | (6%) | |
| Intermediate | 606 | (57%) | 294 | (67%) | 219 | (50%) | 93 | (50%) | |
| High | 362 | (34%) | 101 | (23%) | 180 | (41%) | 81 | (44%) | |
| **Physical activity [1,2]** | **mean** | **(95% CI)** | **mean** | **(95% CI)** | **mean** | **(95% CI)** | **mean** | **(95% CI)** | |
| Daily steps on week days (n = 1,053) | 9,126 | (8,919, 9,333) | 8,618 | (8,247, 8,990) | 9,516 | (9,170, 9,862) | 9,409 | (8,895, 9,923) | |
| Daily steps on weekend days (n = 848) | 8,448 | (8,170, 8,725) | 6,909 | (6,390, 7,428) | 9,385 | (8,925, 9,846) | 9,540 | (8,874, 10,206) | |
| Daily minutes of MVPA on week days (n = 1,053) | 61.0 | (59.4, 62.6) | 56.8 | (53.9, 59.7) | 63.3 | (60.6, 66.0) | 65.6 | (61.6, 69.6) | |
| Daily minutes of MVPA on weekend days (n = 848) | 55.5 | (53.2, 57.7) | 45.6 | (41.4, 49.8) | 60.8 | (57.1, 64.5) | 64.1 | (58.8, 69.5) | |

Total number = 1064, data collected 2013–2016.

[1] Daily steps and minutes of MVPA are adjusted for sex, age group, ethnic group and housing group as fixed effects and household as a random effect in a multi-level model.

[2] Differences between Social and both Intermediate and Market-rent groups were statistically significant, p<0.01.

separately with further adjustment for (i) household as a random effect; (ii) sex, age group, ethnicity and aspirational housing group as fixed effects and household as a random effect; (iii) remaining built environment variables. Differences in weekday and weekend physical activity associations with built environment characteristics from the separate models were formally tested using Z tests. Residuals from all models were checked for assumption of normality to confirm the analytic approach as appropriate.

Further models examined two ways in which built environment variables may contribute to socio-economic differences in PA levels. First, to examine the moderating effect of housing group on the association between the built environment and PA levels, interaction terms for housing group and built environment variables were added to the models and Stata post-estimation commands (testparm) were used to assess their statistical significance. Second, to examine whether differences in access to PA facilities across housing groups translate into differences in levels of PA, attenuation in PA outcomes for each housing group was examined following adjustment for each built environment variable.

Sensitivity analyses examined whether associations remained when the sample was restricted to (i) 837 participants who had data on both weekday and weekend days; (ii) 1,029 participants aged 18+. We also examined whether adjustment for self-reported access to a car or van made any material difference to the findings.

All analyses were carried out in Stata v15.1.

## Results

### Analytical sample

Descriptive statistics are shown in Table 1. Females comprised 58% of the analytical sample, which was largely White (48%). Women, older people and those belonging to ethnic minorities were more prevalent among social compared to intermediate and market-rent housing seekers (p<0.001); the socio-demographic characteristics of intermediate and market-rent housing seekers were similar. Participants seeking relocation to social housing were less physically active compared to other participants. They took 8618 (95%CI [8247; 8990]) steps on weekdays and 6909 (95%CI [6390; 7428]) on weekends, when intermediate housing seekers took 9516 (95%CI [9170; 9862]) and 9385 (95%CI [8925; 9846]), and market-rent housing seekers took 9409 (95%CI [8895;9923]) and 9540 (95%CI [8874;10206]), respectively. Social housing seekers resided in less walkable areas (walkability scores: -0.51, 95%CI [-0.70; -0.31]) compared with both intermediate (0.21, 95%CI [-0.05; 0.46]) and market-rent (0.94, 95%CI [0.48;1.40]) housing seekers. They also lived further away from metropolitan parks (median: 2.53km, IQR [1.43;3.71] for social vs 1.90km, IQR [0.95;2.99] for intermediate and 1.85km, IQR [1.04;3.15] for market-rent housing seekers). Though they lived closer to their nearest local park (median 0.62km, IQR [0.38;0.98] for social vs 0.88km, IQR [0.48;1.35] for intermediate and 0.86km, IQR [0.47;1.37] for market-rent housing seekers). They also had poorer accessibility to public transport (only 22.9% have high accessibility compared to 41.3% for intermediate and 43.5% for market-rent housing seekers).

Participants seeking market-rent housing relocation had a similar level of PA to participants seeking intermediate housing both on weekdays and weekends. They also resided in more walkable areas (walkability scores: 0.94, 95%CI [0.48; 1.40] for market-rent vs 0.21 95% CI [-0.05; 0.46] for intermediate housing seekers), but lived at a similar distance to their closest metropolitan, district and local parks, and had similar accessibility to public transport.

### Built environment and PA-levels associations

Associations between built environment factors and the number of daily steps and amount of MVPA taken on weekdays and weekends are presented in Table 2 and displayed graphically in S1 Fig.

**On weekdays.** In models adjusted for sociodemographics (models 2), both daily steps taken and MVPA accumulated were negatively associated with the distance from home to the closest metropolitan park (mean difference in daily steps (-206, 95%CI [-354;-58]) and in daily minutes of MVPA (-1.8, 95%CI [-2.9;-0.5]) per km of distance to the closest metropolitan

**Table 2. Regression estimates for the association between residential built environment variables and physical activity outcomes (daily steps and daily minutes of MVPA) in the London ENABLE study.**

| | | Model 1 [1] | | Model 2 [2] | | Model 3 [3] | | Model 1 [1] | | Model 2 [2] | | Model 3 [3] | |
|---|---|---|---|---|---|---|---|---|---|---|---|---|---|
| | | β | (95% CI) | β | (95% CI) | β | (95% CI) | β | (95% CI) | β | (95% CI) | β | (95% CI) |
| | | **Outcome: daily steps on weekdays (n = 1053)** | | | | | | **Outcome: daily steps on weekend days (n = 848)** | | | | | |
| Walkability | | 52 | (-30, 134) | -5 | (-86, 76) | 38 | (-57, 132) | **255** | **(139, 371)** | **135** | **(28, 242)** | **144** | **(19, 269)** |
| Distance to closest | | | | | | | | | | | | | |
| metropolitan park (km) | | **-295** | **(-446, -143)** | **-206** | **(-354, -58)** | **-264** | **(-422, -107)** | **-363** | **(-582, -144)** | -125 | (-323, 74) | -144 | (-355, 66) |
| district park (km) | | -93 | (-265, 79) | 7 | (-160, 174) | -64 | (-240, 113) | **-370** | **(-617, -124)** | -180 | (-402, 42) | -139 | (-373, 96) |
| local park (km) | | 170 | (-174, 514) | -84 | (-420, 253) | -185 | (-529, 158) | **1,147** | **(669, 1,624)** | **597** | **(161, 1,032)** | **598** | **(155, 1,040)** |
| Accessiblity to public transport | | | | | | | | | | | | | |
| Low | | 442 | (-362, 1,247) | 767 | (-13, 1,546) | **1,186** | **(296, 2,076)** | **-1,352** | **(-2,516, -188)** | -608 | (-1,658, 442) | 60 | (-1,136, 1,256) |
| Intermediate | | **-482** | **(-948, -15)** | -152 | (-609, 305) | 8 | (-487, 503) | **-1,128** | **(-1,804, -453)** | -365 | (-981, 252) | -8 | (-670, 654) |
| High (reference group) | | | | | | | | | | | | | |
| | | **Outcome: daily minutes of MVPA on weekdays (n = 1053)** | | | | | | **Outcome: daily minutes of MVPA on weekend days (n = 848)** | | | | | |
| Walkability | | 0.6 | (0.0, 1.3) | 0.1 | (-0.6, 0.7) | 0.4 | (-0.3, 1.2) | **2.2** | **(1.2, 3.1)** | **1.2** | **(0.3, 2.1)** | **1.5** | **(0.4, 2.5)** |
| Distance to closest | | | | | | | | | | | | | |
| metropolitan park (km) | | **-2.5** | **(-3.7, -1.3)** | **-1.8** | **(-2.9, -0.6)** | **-2.2** | **(-3.4, -0.9)** | **-3.0** | **(-4.7, -1.2)** | -1.1 | (-2.7, 0.5) | -1.2 | (-2.9, 0.5) |
| district park (km) | | -0.6 | (-2.0, 0.7) | 0.3 | (-1.0, 1.6) | -0.2 | (-1.6, 1.2) | **-2.4** | **(-4.4, -0.4)** | -0.9 | (-2.7, 0.9) | -0.5 | (-2.3, 1.4) |
| local park (km) | | 2.0 | (-0.8, 4.7) | -0.3 | (-3.0, 2.3) | -1.0 | (-3.7, 1.6) | **9.0** | **(5.2, 12.9)** | **4.7** | **(1.2, 8.2)** | **4.9** | **(1.3, 8.5)** |
| Accessiblity to public transport | | | | | | | | | | | | | |
| Low | | 2.7 | (-3.7, 9.1) | 5.8 | (-0.3, 11.9) | **9.7** | **(2.7, 16.6)** | **-9.5** | **(-18.9, -0.1)** | -4.0 | (-12.5, 4.5) | 2.7 | (-7.0, 12.4) |
| Intermediate | | **-4.5** | **(-8.2, -0.8)** | -1.7 | (-5.3, 1.9) | -0.2 | (-4.1, 3.7) | **-8.2** | **(-13.7, -2.8)** | -2.7 | (-7.6, 2.3) | 0.7 | (-4.7, 6.0) |
| High (reference group) | | | | | | | | | | | | | |

Total number = 1064, data collected 2013–2016. Effect estimates highlighted in **bold** are statistically significant, p<0.05.

[1] Model 1 adjusts for household as a random effect to allow for clustering at the household level (referred to as "minimally adjusted model" in the text).

[2] Model 2 additionally adjusts for sex, age group, ethnic group, aspirational housing group as fixed effects.

[3] Model 3 additionally adjusts for all residential built environment variables as fixed effects.

park), indicating that the more distant the metropolitan park, the smaller the number of steps taken and the shorter the time spent in MVPA (Fig 2). Associations between accessibility to public transport and daily steps and MVPA were borderline statistically significant in models adjusted for sociodemographics (mean differences in daily steps (767, 95%CI -13,1546) and in daily minutes of MVPA (5.8, 95%CI [-0.3; 11.9]) for those with a low accessibility compared to those with high accessibility), and reached statistical significance after further adjustment for the other residential built environmental factors (models 3) (mean differences in daily steps: 1186, 95%CI [296;2076], mean difference in daily minutes of MVPA: 9.7, 95%CI [2.7;16.6]). Although adjustment for other residential built environmental factors led to an inflation of the regression estimate and a widening of the confidence interval compared with models adjusting only for sociodemographics, Variance Inflation Factor (VIF) values did not exceed 2, down-playing concerns over multicollinearity.

**On weekend days.** In models adjusted for sociodemographics (models 2), daily steps and MVPA were positively associated with residential walkability (mean difference in daily steps and daily minutes of MVPA per unit of walkability: 135, 95%CI [28; 242], and 1.2, 95%CI [0.3; 2.1], respectively). Daily steps and MVPA also showed a positive association with the distance from home to the closest local park (mean difference in daily steps and daily minutes of MVPA per km of distance to the closest local park: 597, 95%CI [161; 1032], and 4.7, 95%CI

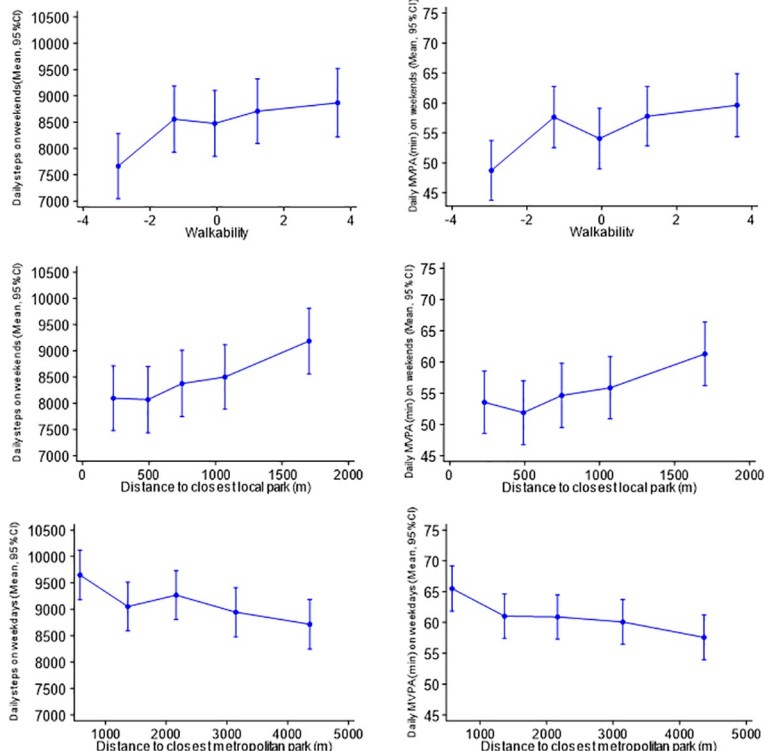

**Fig 2. Daily steps and MVPA on weekends and on weekdays plotted against quintile of residential built environmental factors.** Mean (95%CI) daily steps and mean (95%CI) daily MVPA (min) on weekdays (n = 1,053) and on weekends (n = 848) plotted against A) walkability scores in the residential area at each quintile (median value), B) distance to the closest local park at each quintile (median value), C) distance to the closest district park at each quintile (median value), D) distance to the closest metropolitan park at each quintile (median value), E) accessibility to public transport. Means are adjusted for sex, age group, ethnic group, aspirational housing tenure and clustering at household level.

[1.2; 8.2], respectively). This suggests that the more walkable the residential environment the higher the number of steps taken and the greater the time spent in MVPA, and that the further the closest local park, the higher the number of steps taken and the greater the time spent in MVPA *on weekend days* (Fig 2). Associations remained consistent after further adjustment for the other residential built environmental factors (models 3). Although in models adjusted only for household clustering (models 1) daily steps and MVPA were negatively associated with the distances to the nearest metropolitan and district parks, and with accessibility to public transport, these associations failed to reach significance after adjustment for sociodemographic factors (models 2).

**Comparison of weekday versus weekend day effects.**   There was formal evidence of a difference between weekday and weekend day levels of physical activity in sociodemographic models (Model 2), for (1) increased steps on a weekend day compared with weekday per unit increase in walkability (p = 0.041), (2) increased steps and MVPA at weekends versus weekdays per km increase in distance to local park (p = 0.015, p = 0.026 respectively), and (3) increased steps on a weekday compared with a weekend day associated with lower public transport accessibility (p = 0.02). All other differences between weekend versus weekday physical activity associations with the built environment were not formally statistically significant (p>0.05).

## Pathway 1: Moderating effect of aspirational housing group on the association between the built environment and PA levels

For mean daily steps on weekday, an interaction between low accessibility to public transport and social vs intermediate housing seekers was observed (p = 0.042). However, a post-estimation Wald test suggested the interaction term did not add significantly to the model (p = 0.06), indicating that the residential built environment relates to PA behaviours in a similar way across the three housing groups.

## Pathway 2: Attenuation of the associations between aspirational housing tenure and PA levels by built environment factors

To investigate the degree to which the residential built environment could explain socio-economic differences in PA, we tested the association between aspirational housing tenure and PA levels using various models of built environment factor adjustment (Table 3). Adjustment for walkability decreased differences in daily steps and MVPA *on weekend days* between social and intermediate housing seekers by 1.5% and 2.6%, respectively (i.e. decrease of 36 steps and 0.4min accumulated in MVPA in the gap between social and intermediate housing seekers), adjustment for distance to the closest local park decreased differences in daily steps and MVPA on weekend days between social and intermediate housing seekers by 4.7% and 5.9%, respectively (i.e. decrease of 112 steps and 0.9min accumulated in MVPA), and adjustment for distance to the nearest metropolitan park reduced differences in steps taken *on weekdays* between social and intermediate housing seekers by 9.9% and 11.9%, respectively (i.e. decrease

**Table 3. Effect of adjustment for residential built environment factors in the differences in daily physical activity by housing group in the London-ENABLE study.**

| | | | | Base model with further adjustment for | | | | | | | | | | |
|---|---|---|---|---|---|---|---|---|---|---|---|---|---|---|
| | | Base model [1] | | Walkability | | Distance to metropolitan park | | Distance to district park | | Distance to local park | | Accessibility to public transport | | All BE variables | |
| | Housing group | β | (95% CI) | β | (95% CI) | β | (95% CI) | β | (95% CI) | β | (95% CI) | β | (95% CI) | β | (95% CI) |
| **Daily steps** | | | | | | | | | | | | | | | |
| Weekday, n = 1053 | Social (reference group) | | | | | | | | | | | | | | |
| | Intermediate | 898 | (348, 1448) | 900 | (349, 1451) | 817 | (266, 1369) | 900 | (347, 1452) | 914 | (360, 1469) | 884 | (330, 1,438) | 814 | (255, 1,373) |
| | Market-rent | 791 | (119, 1463) | 797 | (118, 1476) | 723 | (51, 394) | 792 | (119, 1466) | 808 | (132, 1483) | 798 | (122, 1,473) | 727 | (45, 1,409) |
| Weekend n = 848 | Social (reference group) | | | | | | | | | | | | | | |
| | Intermediate | 2477 | (1721, 3232) | 2441 | (1689, 3194) | 2436 | (1678, 3194) | 2427 | (1671, 3184) | 2365 | (1607, 3122) | 2409 | (1646, 3172) | 2242 | (1477, 3007) |
| | Market-rent | 2631 | (1737, 3526) | 2474 | (1575, 3373) | 2598 | (1702, 3494) | 2596 | (1702, 3490) | 2513 | (1618, 3409) | 2545 | (1642, 3448) | 2280 | (1375, 3186) |
| **Daily minutes of MVPA** | | | | | | | | | | | | | | | |
| Weekday n = 1053 | Social (reference group) | | | | | | | | | | | | | | |
| | Intermediate | 6.6 | (2.3, 10.9) | 6.6 | (2.3, 10.9) | 5.9 | (1.6, 10.2) | 6.7 | (2.4, 11.0) | 6.7 | (2.3, 11.0) | 6.4 | (2.1, 10.7) | 5.8 | (1.5, 10.2) |
| | Market-rent | 8.9 | (3.6, 14.1) | 8.8 | (3.5, 14.1) | 8.3 | (3.0, 13.5) | 8.9 | (3.7, 14.2) | 8.9 | (3.7, 14.2) | 8.8 | (3.6, 14.1) | 8.1 | (2.8, 13.5) |
| Weekend n = 848 | Social (reference group) | | | | | | | | | | | | | | |
| | Intermediate | 15.3 | (9.2, 21.4) | 14.9 | (8.9, 21.0) | 14.9 | (8.8, 21.0) | 15.0 | (8.9, 21.2) | 14.4 | (8.3, 20.5) | 14.8 | (8.6, 20.9) | 13.6 | (7.4, 19.8) |
| | Market-rent | 18.6 | (11.4, 25.8) | 17.2 | (9.9, 24.4) | 18.3 | (11.0, 25.5) | 18.4 | (11.2, 25.7) | 17.7 | (10.4, 24.9) | 18.0 | (10.7, 25.3) | 15.8 | (8.4, 23.1) |

[1] The base model is adjusted for sex, age group, ethnic group as fixed effects and household as a random effect to allow for clustering in a multi-level model.

of 81 steps and 0.7min accumulated in MVPA). These findings suggest that aspirational housing tenure differences in PA relate to the differences in neighbourhood walkability and distance to local and metropolitan parks observed across the three housing groups.

Sensitivity analyses performed on the subsample of 837 participants with PA data on both weekdays and weekends showed consistency with the main findings presented above (S5–S7 Tables). Analyses performed on the subsample of 1,029 participants aged 18+ year (i.e. excluding 35 participants) and with adjustment for self-reported access to a car or van (in 460/1,064) also showed similar associations (results not shown).

## Discussion

This study aimed to measure the association of a range of residential built environment factors with PA levels on weekdays and weekend days. It further explored whether the residential built environment contributes to socio-economic differences in PA.

In step with our first aim, we found that participants from more walkable neighbourhoods took more steps and accumulated more MVPA on weekends compared to those from less walkable neighbourhoods. These findings add to the growing evidence, both worldwide [45, 46] and in the UK [22], that the more walkable an urban environment, the more attractive for active modes of travel. For instance, Stockton *et al.*, using data from Londoners aged 35 to 55 years from the Whitehall II study, similarly found that those residing in the most walkable neighbourhoods spent significantly more time walking than those in the least walkable ones [22]. Our findings however highlighted a stronger association with neighbourhood walkability on weekends. Research has shown that the time spent in the residential neighbourhood modifies the association between neighbourhood walkability and PA levels, with the longer the time spent, the stronger the relationship [47]. Our findings of a stronger relationship on weekends could suggest that ENABLE London participants spent more time in the home vicinity on weekends. This interpretation is consistent with a UK time survey highlighting that people spend more time on leisure and domestic work on weekends [29].

We also found evidence of association between park proximity and PA levels. Increased number of steps and amount of MVPA were associated with increased distance to the nearest local park on weekends, but with decreased distance to the nearest metropolitan park on weekdays. Local parks are on average within walking distance from home [37], rendering plausible the assumption of physically active commutes. Local parks are also small, and provided with none or few sport facilities [37]. The journey to rather than activities within local parks may therefore contribute to PA levels. This would explain why the number of steps and amount of MVPA increase with distance to local parks. Conversely, metropolitan parks are often situated beyond the commonly admitted walking distance. Interest in reaching them may therefore decrease with increased distance. Metropolitan parks are also large and well provided with sport facilities [37]. Increased number of sport facilities within a park has been associated with increased use of that park for PA [48]. Activities undertaken within metropolitan parks, rather than the journey to reach them, may therefore contribute to PA levels. This would explain that the number of steps and amount of MVPA increase when distance to the park decreases. Like others highlighted [49], park characteristics such as distance, size and enclosed facilities are important to consider when exploring the role of greenspaces on PA levels. Overlooking these aspects may help explain inconsistent findings in the literature on access to greenspaces and PA, which reports both null [25, 35] and beneficial [23, 24, 26, 49–51] associations. Interestingly, we found that distance to local parks related to PA levels only on weekends, whilst distance to metropolitan parks only on weekdays. Local parks are common features of the immediate home vicinity, which individuals may make greater use on the weekend, when they

are assumed to spend more time at home. Because metropolitan parks are further from the home, their use may be associated with other weekdays activities, like work, which often takes individuals beyond their local neighbourhood. These assumptions would require further testing.

Positive associations between accessibility to public transport and PA levels have been found worldwide [52], including in the UK [23], suggesting that interest in use of public transport may decrease with decreasing accessibility. For instance, Sallis *et al.* found that accessibility to public transport, measured as the density of public transport access points within a 1-km buffer centred on the home address, positively relates to adult PA levels in 14 cities worldwide, including Stoke-on-Trent, UK [23]. Yet, our findings did not fully align with this conclusion, as the positive association between accessibility to public transport and steps and MVPA found on weekends failed to reach significance after adjustment for sociodemographic factors. More surprisingly, the association becomes negative on weekdays, with participants experiencing a low accessibility to public transport taking more daily steps compared to those with high accessibility. To our knowledge, such findings have not been reported in the literature and raise several questions. Participant's heavy reliance on public transportation for inescapable weekday activities, like work, may explain why those with poor accessibility to public transport walk more on weekdays than those who live closer. Examining whether this negative association between public transport accessibility and PA on weekdays holds for individuals who have alternative ways of travelling (e.g., car) offers an avenue for further research. On a different note, our measure of public transport accessibility is not strictly comparable to density-based measures (simple count of public transport stops within a given area) used in other studies, in that it also accounts for frequency and reliability of service. Yet, our variable reflects a concept of public transport accessibility not that different to more basic density-based variables and is unlikely to be responsible for inverting the direction of the association. Overall, opposite directions of the associations on weekend and weekdays may suggest that the use of public transport may depend on utility and travel function: an inescapable commute to work on weekdays and leisure and recreation on the weekends.

Put altogether, our findings suggest that the residential built environment is associated with PA, but that this varies according to day of the week. Features in the immediate vicinity of the home, like local parks and walkability captured within 1-km home-centred buffer, were found to be associated with PA on the weekend. Conversely, features like metropolitan parks that were beyond the immediate home vicinity, and public transportation that take individuals beyond the limit of their residential neighbourhood, were found to relate to PA levels on weekdays. The intertwinement of spatial and temporal patterns of human activities unveiled here suggests that PA behaviours are shaped by broad contextual forces that go beyond the simple assessment of what is physically accessible from home [53]. Others have highlighted that space *and* time constraints in adults' daily activities are important factors that shape the impact of residential neighbourhood attributes on PA. For instance, Cerin *et al.* found that time of the day and day of week were significant moderators of residential built environment-MVPA associations [26]. The combined use of GPS and accelerometer data appears as a promising avenue for research to deepen our understanding of the spatio-temporal determinants of PA behaviours [54]. Such combined data would, for instance, have allowed us to confirm/overturn our assumptions that the association between metropolitan parks and PA on weekdays owe to PA done *within* the metropolitan parks. By identifying the activities that precede and follow a visit to the metropolitan park, we could further have possibly identified some time specific constraints that encourage individuals to use metropolitan parks on weekdays rather than weekends.

Finally, in line with our second aim, we explored two plausible pathways by which the built environment may contribute to socio-economic differences in PA: (i) differences in the way

the housing groups interact with the features supportive of active behaviours in the residential environment, and (ii) differences in the availability of these features across the housing tenure groups. We found little evidence for the former pathway, with an absence of interactions between residential built environment factors and housing tenure group in relation to PA levels on weekdays and weekends. This suggests that the residential built environment relates to PA levels in a similar way across the three socio-economic groups, regardless of whether PA occurs on weekdays or weekends. This is in line with some studies [55], but disagrees with others like Jones et al., who found a strong park distance decay effect on park use for the most affluent neighbourhoods, but not for the most deprived ones in Bristol, UK [56]. As for the second pathway, we found that social housing seekers lived in less walkable environments, closer to local parks and further away from metropolitan parks compared to intermediate and market-rent housing seekers. In turn, decreased neighbourhood walkability, shorter distance to local parks, and increased distance to metropolitan parks were all associated with a decreased number of daily steps and time spent in MVPA. These findings align with the idea that disadvantaged individuals are distributed in neighbourhoods with less health-promoting facilities [16]. There is mounting evidence from various countries of socio-economic disparities in the distribution of PA-promoting features in the residential neighbourhood [57, 58], and in PA levels [5–7]. Yet, comparatively, there is little evidence of a link between these two disparities. We found that the association between socio-economic status and PA behaviours on weekdays was attenuated by disparities in distance to metropolitan parks (decreased differences in daily steps and MVPA between social and intermediate housing seekers of 9.9% and 11.9%, respectively). The associations between socio-economic status and PA behaviours on weekends were also slightly attenuated by disparities in neighbourhood walkability (1.5% and 2.6% for steps and MVPA, respectively) and distance to local parks (4.7% and 5.9% for steps and MVPA, respectively). These findings support the notion that disadvantaged people are less physically active because they live into less health-promoting neighbourhoods [41, 59]. Only few empirical studies have so far provided evidence that the residential built environment contributes to socio-economic differences in PA. And like ours, these studies found that only a limited amount of the socio-economic differences in PA relate to differences in environmental features in the residential area [14, 60]. More research is needed to confirm the weight of the residential built environment on socio-economic differences in PA.

## Strengths and limitations

Strengths of this paper include the use of validated objective measures of PA [61], assessing weekday/weekend variations in PA, and exploring the contribution of the residential built environment in explaining socio-economic differences in PA levels. Sensitivity analyses further strengthened our findings by showing consistent results in both our inclusive analytical sample and the more restricted sample of participants with PA data on *both* weekdays *and* weekends, and participants 18 years or above. Durations of adequate wear-time to define habitual activity vary. Previous studies have often used a minimum of 3 to 5 days of accelerometry data [62], with as little as 480 minutes [63] per day to 600 minutes [64] per day of daily recording. We chose to include all days of ≥540 minutes of registered accelerometry time during at least one day to lessen attrition bias and maximise inclusion of hard to reach groups, i.e., those from social housing who are likely to have lower compliance and record less PA data. This decision was part of the study design and was in our a priori Statistical Analysis Plan approved by the ENABLE London Steering Committee [30]. There are several limitations which warrant consideration. This study draws on the baseline sample of a natural experiment, where recruits were deemed eligible if they were seeking to relocate to East Village. Such

sampling strategy limits external validity, as the participants may not be truly representative of the London population as a whole. Moreover, the cross-sectional study design is prone to selection bias (i.e., those living in less walkable neighbourhoods are intrinsically different to those who do not), and therefore restricts interpretations about direction of effects.

Moreover, partly because we overlooked non-residential exposures to the built environment, we were not able to align with calls for addressing the Uncertain Geographic Context Problem (i.e., the extent to which areal units of measurement deviate from the geographic context truly experienced by individuals) [53]. Although the spatial unit used to derive our environmental exposure complies with the wide majority of studies that have looked at the way the residential built environment relates to health behaviours (i.e. using an home-centred buffer within a walkable distance radius), focusing on the residential neighbourhood only may have led to misestimation of the association between the residential environment and health behaviours [65].

## Conclusions

This study adds to the evidence that the residential built environment could act as a potential lever for increasing physical activity levels in adults. Interestingly, individuals from lower socio-economic status may disproportionately benefit from greater availability of physical activity resources, because they tend to reside in urban environment less supportive of PA. Policymakers should however be sensitive to that possibility that interventions might have different impacts on physical activity on weekends and weekdays. More evidence, drawn from longitudinal studies and relying on the combined use of GPS and accelerometer data is needed though, to confirm the direction and interpretation of these associations.

## Supporting information

**S1 Table. Sociodemographics of ENABLE London participants included in (n = 1064) and excluded from (n = 214) the analyses.**
(DOCX)

**S2 Table. Description, number and size of metropolitan, district and local parks.**
(DOCX)

**S3 Table. Data sources and versions used for computing residential built environment variables.**
(DOCX)

**S4 Table. Criteria for allocation of housing tenures in East Village.**
(DOCX)

**S5 Table. Baseline demographic and daily steps of 837 ENABLE-London participants who provided daily physical activity data for weekdays and weekend days.**
(DOCX)

**S6 Table. Regression estimates for the association between residential built environment variables and physical activity outcomes (daily steps and daily minutes of MVPA) in the London ENABLE study for 837 participants who provided PA data on weekdays and weekend days.**
(DOCX)

**S7 Table. Effect of adjustment for residential built environment factors in the differences in daily physical activity by housing group in the London-ENABLE study in 837**

**participants who provided PA data on both weekdays and weekend days.**
(DOCX)

**S1 Fig. Daily steps and MVPA on weekends and on weekdays plotted against quintile of residential built environmental factors.** Legend: Mean (95%CI) daily steps and mean (95% CI) daily MVPA (min) on weekdays (n = 1,053) and on weekend days (n = 848) plotted against A) walkability scores in the residential area at each quintile (median value), B) distance to the closest local park at each quintile (median value), C) distance to the closest district park at each quintile (median value), D) distance to the closest metropolitan park at each quintile (median value), E) accessibility to public transport. Means are adjusted for sex, age group, ethnic group, aspirational housing tenure and clustering at household level.
(DOCX)

## Acknowledgments

The authors would like to thank the East Thames Group, Triathlon Homes and Get Living London who have assisted in recruiting participants into the ENABLE London study. The ENABLE London study was advised by a Steering Committee chaired by Professor Hazel Inskip (University of Southampton), with Dr David Ogilvie (University of Cambridge) and Professor Andy Jones (University of East Anglia) as academic advisors and Mrs Kate Worley (formerly East Thames Group Assistant Director for Strategic Housing) as the lay/stakeholder member. The authors are grateful to the members of the ENABLE London study team (in particular Aine Hogan, Katrin Peuker, Cathy McKay) and to participating households. Residential built environment variables have been created using material from GiGL and its partners, TfL, and Ordnance Survey (Ordnance Survey © Crown Copyright 2018. All rights reserved. License number 100034829).

## Author Contributions

**Conceptualization:** Christelle Clary, Daniel Lewis, Alicja R. Rudnicka, Duncan Procter, Angie S. Page, Ashley R. Cooper, Anne Ellaway, Billie Giles-Corti, Peter H. Whincup, Derek G. Cook, Christopher G. Owen, Steven Cummins.

**Data curation:** Elizabeth S. Limb, Claire M. Nightingale, Christopher G. Owen.

**Formal analysis:** Christelle Clary, Elizabeth S. Limb, Claire M. Nightingale.

**Funding acquisition:** Alicja R. Rudnicka, Angie S. Page, Ashley R. Cooper, Anne Ellaway, Billie Giles-Corti, Peter H. Whincup, Derek G. Cook, Christopher G. Owen, Steven Cummins.

**Investigation:** Bina Ram, Christopher G. Owen.

**Methodology:** Christelle Clary, Daniel Lewis, Elizabeth S. Limb, Claire M. Nightingale, Alicja R. Rudnicka, Duncan Procter, Angie S. Page, Ashley R. Cooper, Derek G. Cook, Christopher G. Owen, Steven Cummins.

**Project administration:** Bina Ram, Christopher G. Owen.

**Supervision:** Daniel Lewis, Christopher G. Owen.

**Writing – original draft:** Christelle Clary.

**Writing – review & editing:** Christelle Clary, Daniel Lewis, Elizabeth S. Limb, Claire M. Nightingale, Bina Ram, Alicja R. Rudnicka, Duncan Procter, Angie S. Page, Ashley R.

Cooper, Anne Ellaway, Billie Giles-Corti, Peter H. Whincup, Derek G. Cook, Christopher G. Owen, Steven Cummins.

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
