## [Decision Letter · Decision Letter 0]

16 Apr 2020

PONE-D-19-33687

Weekend and weekday differences in associations between the residential built environment and physical activity: findings from the ENABLE-London Study

PLOS ONE

Dear Prof. Owen,

Thank you for submitting your manuscript to PLOS ONE. After careful consideration, we feel that it has merit but does not fully meet PLOS ONE’s publication criteria as it currently stands. Therefore, we invite you to submit a revised version of the manuscript that addresses the points raised during the review process.

We would appreciate receiving your revised manuscript by May 31 2020 11:59PM. To enhance the reproducibility of your results, we recommend that if applicable you deposit your laboratory protocols in protocols.io, where a protocol can be assigned its own identifier (DOI) such that it can be cited independently in the future. For instructions see: http://journals.plos.org/plosone/s/submission-guidelines#loc-laboratory-protocols

We look forward to receiving your revised manuscript.

Kind regards,

Adewale L. Oyeyemi, Ph.D

Academic Editor

PLOS ONE

Journal Requirements:

2. Thank you for including your ethics statement: Full ethical approval was obtained from the relevant Multi-Centre Research Ethics Committee (REC Reference 12/LO/1031). All participants provided written informed consent.

Reviewers' comments:

Reviewer's Responses to Questions

**Comments to the Author**

1. Is the manuscript technically sound, and do the data support the conclusions?

Reviewer #1: Partly

2. Has the statistical analysis been performed appropriately and rigorously? 

Reviewer #1: Yes

3. Have the authors made all data underlying the findings in their manuscript fully available?

Reviewer #1: No

4. Is the manuscript presented in an intelligible fashion and written in standard English?

Reviewer #1: Yes

5. Review Comments to the Author

Reviewer #1: Thank you for the opportunity of reviewing this interesting study. I hope these comments and suggestions help strengthen this manuscript.

This study aimed to determine the association of the residential built environment with physical activity outcomes (daily steps and daily MVPA), stratifying by type of day (weekend vs. weekdays). Next, authors explored potential effect measure modification of these relations by socioeconomic status.

Overall strengths are a large sample size, objective measures for physical activity, and a decent response rate for these types of studies. The authors explore important research questions (the difference in the effect of built environment variables on physical activity outcomes by type of week-day, and the role of the BE in the SES-PA relationship).

INTRODUCTION

• There is a mention of the prevalence of physical inactivity in the UK based on old recommendations of accumulating at least 150 mins/wk of MVPA within 10-minute bouts. Please cite evidence (prevalence of inactivity) based on current guidelines, without the stipulation of the 10-minute bouts.

• Regarding the phrase: “recently, epidemiological research has increasingly incorporated socio-ecological models that acknowledge the role of the built environment, especially the local residential environment, in determining PA behaviors”. Please change the term “determinants” to “influencing”, or, state “as potential determinants of PA behaviors”. The term “determinants” implies strong evidence of causality, which for the most part is still lacking for the role of built environments on health behaviors.

• In the first paragraph, the authors seem to make a case about findings being mixed/inconclusive with regards to the role of the built environment on socioeconomic disparities. They then cite multiple studies with divergent findings. To me, it is apparent that more than having mixed findings, we have evidence of context-specific differences. In some places, low-income neighborhoods are deprived of adequate BE resources for PA. In other settings, it is precisely in low-income neighborhoods where one finds the most BE resources for PA (and, because virtually all evidence is from cross-sectional studies, this might be as a response from local governments to equalize the playing field for these economically disadvantaged areas). This could be much better framed acknowledging the role of context in these relations.

METHODS

Sampling

• This is the baseline sample of a natural experiment, where recruits were deemed eligible if they were people seeking to relocate to a new residence in the short term. I believe a more extensive comment on the limitation of such a sampling strategy for the aims of this specific analysis is warranted. This isn’t a truly representative sample of the population, yet the data are being treated as though this were a typical cross-sectional survey. While the sample serves the purpose of the parent study (a natural experiment) well, there are limitations to conducting a cross-sectional analysis with its baseline data, mainly due to selection bias and external validity limitations, which require further acknowledgement.

Variables

• Accelerometer-derived PA outcomes: when describing the outcome variables, authors state that daily steps were “adjusted for day of the week, day order of recording and month of data collection”. I assume this is referring to the regression models per se? Or, are the authors implying that the actual variables are somehow weighted for these characteristics? This is not typically seen in accelerometry studies, and I believe requires further explanation. If this is something done at the modeling step, I recommend removing this information from the variable description section, as this can confuse the reader, and simply include it in the modeling section.

• MVPA variable: The use of a single day of valid accelerometry data (and of only 9 hours) certainly seems a bit low. Usual standards call for a minimum of 4 days with at least 10 hours of valid data per day. However, authors justify this by referring to a sensitivity analysis performed against complete data, which they define as that from participants with at least 4 days of 9 hours of data or more (instead of 10 hours). A single day of valid accelerometry data is thought to be sufficient for surveillance of populations (when truly representative samples are used), but I do worry that for this dataset, which is composed of a non-representative sample of London residents, and is not done for surveillance purposes, this may be a bit of a stretch.

• Land-use mix and net residential density variables: can the authors please provide further detail on the basis for the log-transformation of these variables? I haven’t seen this done before in other studies from other global settings. They state it is to “fit a comparable scale”. Comparable to what?

• When explaining the rationale for the 1-KM network buffers, it might be good to state the average walking distance/time this represents (e.g., about a 10-15 minute walk from home in any direction for most people).

• Proximity to parks variable: other studies have found limited utility in these distance-based variable, including IPEN-adult, a 12 country study, where park density within 1-KM buffers was deemed a more useful predictor of physical activity. Did the authors consider using a buffer-based variable for parks too? Why was this discarded?

• I find it very interesting that car ownership was not deemed as a relevant confounder in this analysis, as this contradicts what we know about this variable in other parts of the world.

Statistical Analysis

• The first phrase is not clear. Authors mention a series of variables “were examined”. This could mean anything. Please be clear on the actual analytic procedures applied for “examining” these variables.

• Was a test for interaction actually performed to determine of day of the week (weekend vs. weekday) actually modifies the effect observed for some of these BE variables on the PA outcomes?

• The use of multilevel models seems adequate, but I am surprised at such a high ICC (0.3, wow!). It would be important to report what percentage of the sample shared a household with another sample participant. Most studies of this type only accept one participant per household to avoid this level of clustering. Also, did the authors explore and, if needed, account for clustering at the neighborhood level (e.g., people residing in the same zip-code or small administrative geospatial unit?).

• Could the authors explicitly state their hypotheses for the two different type of analyses proposed for examining the role of the BE in explaining PA-SES differences? I can follow the rational for the effect-measure modification analysis (testing for interaction terms of BE X SES variables). The simply adjustment by BE variables of the effect of SES on PA seems a repetition from the first modeling step, and implies confounding vs. effect measure modification, when, per the introduction, it seems that the authors have an effect modification hypothesis to begin with.

• Also, the authors don’t mention which test for interaction and alpha level they used in their analysis (hopefully this was not just based on the p-value of the interaction term in the regression model – but was the result of a formal test for interaction – please clarify).

• Outcome variables in models (steps and mins of MVPA/day): were these transformed in any way for achieving normality? Physical activity data are usually not normally distributed, hence the use of linear regression models usually requires a data transformation step. Alternatively, one could consider the use of logistic regression models (multinomial, politomous, or binary).

• Did the authors examine potential multicollinearity in their models? This is usually a concern with some built environment variables in a single model. Please outline the methods via which this was examined.

RESULTS

• Table 2 title: Should read: “Regression estimates for the association between residential built environment favors with physical activity outcomes (daily steps and mins/wk of MVPA), among… XXX [insert study name, year of data collection, etc.)”. It is customary to list the independent variable first and the dependent variable last, not the other way around.

• For all table/figure titles: add information on the study and sample, and year data were collected, as tables and figures should be “stand-alone” items. Figure 2 is very nice and helpful for the reader.

• I like the way the results for the examination of the role of the BE on the PA-SES relation are presented by two potential pathways. However, it would have been much more helpful to lay out the two hypothesized pathways since the introduction and methods (Statistical analysis) section, so the readers understand why both approaches (test for effect modification vs. BE as a confounder of the SES-PA relation) were explored. Otherwise the “two pathways” seem more a bit of an afterthought. For this purpose, it might be good for the authors to present a figure with the hypothetical DAGs of the two pathways they are exploring, a priori (i.e., before the results section).

DISCUSSION

• I enjoyed reading the interpretation of the results regarding the counter-intuitive finding on longer distance to small parks being associated to higher levels of MVPA on weekends. This makes a lot of sense and it is not something the readers could have derived just by seeing the estimates.

• The discussion on the public transit results should be re-framed. I would suggest not opening with a statement on how they found a positive association with access to transit and PA, given that in reality once the models were adjusted this was not significant (the authors do state this, but I just don’t see the need to open with a statement of a positive relation). I think a deeper discussion is warranted to explain why their findings are inconsistent with most other reports of transit and PA. It is counterintuitive that those with lower availability to public transit have higher levels of PA, in a model that is also adjusted for SES. I wonder if adjusting for car ownership would have made a difference for this specific variable [access to transit] (?). The discussion on this topic should also include some reflection on the difference in measures of access to transit. Most other studies have relied on transit stop density measures (count of public transit stops within a buffer), and thus, are not truly comparable to the measure used in this study.

• Limitations paragraph:

o The use of a single day of accelerometry data with at least 9 hours is not in line with previous studies and should be acknowledged.

o The transit access measure, although apparently more sophisticated than what others use, is not comparable to what most researchers use when examining the role of transit infrastructure on PA.

o The Uncertain Geographic Context Problem and this study’s inability to address it should be listed as a limitation.

o Another major limitation is the authors simply assumed differences in the relation of the BE and PA by weekday (effect modification by type of weekday), which is reasonable, but a stronger study analysis would have included a confirmation of this with a test for interaction, followed by the stratified analysis.

o Finally, selection bias due to a) the response rate, and b) the eligibility criteria (this is not a representative sample of London residents, rather, it targets people with a desire to move) needs to be fully disclosed, with a description of how these factors are likely limiting the external validity of the study. In particular, because a reason for moving might be the BE features of the new neighborhood where people are hoping to move to.

6. PLOS authors have the option to publish the peer review history of their article (what does this mean?). If published, this will include your full peer review and any attached files.

Reviewer #1: Yes: Deborah Salvo

---

## [Author Response · Author response to Decision Letter 0]

8 Jun 2020

Further Editorial queries (dated 28th May 2020)

E1. Thank you for including your ethics statement: Full ethical approval was obtained from the relevant Multi-Centre Research Ethics Committee (REC Reference 12/LO/1031). All participants provided written informed consent.

Response to Editorial comments

We have amended the ethics statement in both the paper and submission form to indicate that full ethical approval was obtained from the City Road and Hampstead Ethical Review Board (12/LO/1031).

E2. Are ethical or legal restrictions on sharing a de-identified data set, please explain them in detail (e.g., data contain potentially identifying or sensitive patient information) and who has imposed them (e.g., an ethics committee). Please also provide contact information for a data access committee, ethics committee, or other institutional body to which data requests may be sent.

Response to Editorial comments

While we are fully supportive of providing anonymised data, unfortunately this was not part of the original ethics approval (see response to E1 above), and permission to share anonymised data was not included in the ENABLE London participant consent form. This raises issues with General Data Protection Regulations. This is why we have said that data from this study are available upon request, as we hope these would be considered by our data protection team on an individual basis.

Reviewer 1

1. Thank you for the opportunity of reviewing this interesting study. I hope these comments and suggestions help strengthen this manuscript.

This study aimed to determine the association of the residential built environment with physical activity outcomes (daily steps and daily MVPA), stratifying by type of day (weekend vs. weekdays). Next, authors explored potential effect measure modification of these relations by socioeconomic status.

Overall strengths are a large sample size, objective measures for physical activity, and a decent response rate for these types of studies. The authors explore important research questions (the difference in the effect of built environment variables on physical activity outcomes by type of week-day, and the role of the BE in the SES-PA relationship).

Response to reviewer

Thank you for these positive comments about our work, and for all the suggestions that have helped us strengthen the manuscript.

2. Introduction

There is a mention of the prevalence of physical inactivity in the UK based on old recommendations of accumulating at least 150 mins/wk of MVPA within 10-minute bouts. Please cite evidence (prevalence of inactivity) based on current guidelines, without the stipulation of the 10-minute bouts.

Response to reviewer

We thank the reviewer for this suggestion. We have amended the text (page 3, lines 44-46), citing the most recent evidence available (Sport England 2018/19 report published in April 2020), as follows: “Current population levels of PA are too low in the UK, with 37% of adults aged 16 years or more not meeting recommended levels of activity of at least 150 minutes of moderate-intensity per week (1)”.

3. Introduction

Regarding the phrase: “recently, epidemiological research has increasingly incorporated socio-ecological models that acknowledge the role of the built environment, especially the local residential environment, in determining PA behaviors”. Please change the term “determinants” to “influencing”, or, state “as potential determinants of PA behaviors”. The term “determinants” implies strong evidence of causality, which for the most part is still lacking for the role of built environments on health behaviors.

Response to reviewer

As suggested, we have amended the sentence on page 3 (lines 47-50) as follows: “Recently, epidemiological research has increasingly incorporated socio-ecological models that acknowledge the role of the built environment, especially the local residential environment, in influencing PA behaviours”.

4. Introduction

In the first paragraph, the authors seem to make a case about findings being mixed/inconclusive with regards to the role of the built environment on socioeconomic disparities. They then cite multiple studies with divergent findings. To me, it is apparent that more than having mixed findings, we have evidence of context-specific differences. In some places, low-income neighborhoods are deprived of adequate BE resources for PA. In other settings, it is precisely in low-income neighborhoods where one finds the most BE resources for PA (and, because virtually all evidence is from cross-sectional studies, this might be as a response from local governments to equalize the playing field for these economically disadvantaged areas). This could be much better framed acknowledging the role of context in these relations.

Response to reviewer

We agree with this suggestion. We have reframed this introductive paragraph to highlight that mixed findings can reflect diverse local realities, and to enhance the role of context in the BE and PA associations. We have also used this opportunity to clarify the way the built environment contribution to socio-economic differences in PA is explored in the literature. Overall, this introductive paragraph now aligns better with the aims of the study as described at the end of the Introduction and partly addresses the issues raised in the reviewer’s comment #15. The reframed paragraph on page 3 (lines 50-65) now reads:

“The extent to which the residential built environment contributes to individual socio-economic differences in PA has been explored by drawing on two hypothetical pathways: (i) a deprivation-amplification effect (2), whereby disadvantaged individuals are less exposed to health-promoting facilities in their residential neighbourhood, and (ii) a moderating effect, by which socio-economic groups use the physical activity facilities available in their neighbourhood differently (3). Regarding the deprivation-amplification pathway, some UK studies have reported that the most affluent urban areas have the poorest accessibility to recreational PA facilities (4-7), but others the most deprived (8). As for the moderating effect pathway, some studies have reported that the association between the residential built environment and PA was moderated by the socio-economic position (9, 10), whereas others did not find such evidence (11, 12). These mixed findings likely reflect diverse local realities, that may owe to local population specificities or regional policy interventions. More research is needed to depict a comprehensive, overarching view of how built environment, socio-economic status and PA behaviours interrelate, and, in turn, deliver effective ‘contextually sensitive’ policy interventions (13).”.

5. Methods, Sampling

This is the baseline sample of a natural experiment, where recruits were deemed eligible if they were people seeking to relocate to a new residence in the short term. I believe a more extensive comment on the limitation of such a sampling strategy for the aims of this specific analysis is warranted. This isn’t a truly representative sample of the population, yet the data are being treated as though this were a typical cross-sectional survey. While the sample serves the purpose of the parent study (a natural experiment) well, there are limitations to conducting a cross-sectional analysis with its baseline data, mainly due to selection bias and external validity limitations, which require further acknowledgement.

Response to reviewer

We agree with the reviewer that the limitations related to our sampling strategy should be better acknowledged. We have added the following paragraph in the Limitations section of the manuscript (page 22, lines 488-494): “This study draws on the baseline sample of a natural experiment, where recruits were deemed eligible if they were seeking to relocate to East Village. Such sampling strategy limits external validity, as the participants may not be truly representative of the London population as a whole. Moreover, the cross-sectional study design is prone to selection bias (i.e., those living in less walkable neighbourhoods are intrinsically different to those who do not), and therefore restricts interpretations about direction of effects.”.

6. Methods, Variables

Accelerometer-derived PA outcomes: when describing the outcome variables, authors state that daily steps were “adjusted for day of the week, day order of recording and month of data collection”. I assume this is referring to the regression models per se? Or, are the authors implying that the actual variables are somehow weighted for these characteristics? This is not typically seen in accelerometry studies, and I believe requires further explanation. If this is something done at the modeling step, I recommend removing this information from the variable description section, as this can confuse the reader, and simply include it in the modeling section.

Response to reviewer

We agree that this could have been clearer and we have amended the text to clarify different aspects of the accelerometry assessment. Participants were asked to wear the accelerometer for 7 consecutive days from waking to going to bed. This provided daily step counts and daily minutes spent in moderate to vigorous physical activity (MVPA). Days where recorded wear time was less than 540 minutes were dropped from the analysis. Average daily step count and average daily MVPA were obtained using multi-level linear regression models (level 1 was day within individual and level 2 was individual). Daily steps (minutes of MVPA) were regressed on day-order-of-wear, day-of-week and month-of-wear. This method provides unbiased estimates of physical activity levels for each individual, which were used to examine associations with residential built environment variables (as outlined in the Statistical Analysis section). We have amended the text as below to demarcate between derivation of the physical activity variable, and the statistical methods used to relate the residential built environment to these derived measures of physical activity.

In the Methods section, under the ‘Accelerometer-derived PA outcomes’ subheading (page 7, lines 135-142) we have added the following: “Participants were asked to wear a hip-mounted ActiGraph GT3X+ accelerometer for 7 consecutive days during waking hours. This provided daily measures of steps and time spent in moderate-to-vigorous physical activity (MVPA), based on the standard threshold of ≥ 1952 counts per minute (14). We excluded days of accelerometer data where the registered wear time was less than 540 minutes. Multilevel linear regression models were fitted to allow for repeated measurements of PA, level 1 was day within individual and level 2 was individual. Daily steps (minutes of MVPA) were regressed on day-order-of-wear, day-of-week and month-of-wear, and an unbiased estimate of average daily steps (minutes of MVPA) obtained for each participant for weekdays and weekend days separately.”.

In the Statistical Analysis (page 11, lines 222-242) section we have altered the text to be clearer about the multi-level models used, as follows: 

“Multi-level linear regression models were used to examine the association between daily PA (steps and minutes of MVPA) and residential built environment variables (walkability, distance to parks and accessibility to public transport) on weekdays and weekend days separately. Average daily steps (minutes of MVPA) were regressed on each built environment variable separately with further adjustment for (i) household as a random effect; (ii) sex, age group, ethnicity and aspirational housing group as fixed effects and household as a random effect; (iii) remaining built environment variables. Residuals from all models were checked for assumption of normality to confirm the analytic approach as appropriate.

Further models examined two ways in which built environment variables may contribute to socio-economic differences in PA levels. First, to examine the moderating effect of housing group on the association between the built environment and PA levels, interaction terms for housing group and built environment variables were added to the models and Stata post-estimation commands (testparm) were used to assess their statistical significance. Second, to examine whether differences in access to PA facilities across housing groups translate into differences in levels of PA, attenuation in PA outcomes for each housing group was examined following adjustment for each built environment variable. 

Sensitivity analyses examined whether associations remained when the sample was restricted to (i) 837 participants who had data on both weekday and weekend days; (ii) 1,029 participants aged 18+.”.

7. Methods, MVPA variable

The use of a single day of valid accelerometry data (and of only 9 hours) certainly seems a bit low. Usual standards call for a minimum of 4 days with at least 10 hours of valid data per day. However, authors justify this by referring to a sensitivity analysis performed against complete data, which they define as that from participants with at least 4 days of 9 hours of data or more (instead of 10 hours). A single day of valid accelerometry data is thought to be sufficient for surveillance of populations (when truly representative samples are used), but I do worry that for this dataset, which is composed of a non-representative sample of London residents, and is not done for surveillance purposes, this may be a bit of a stretch.

Response to reviewer

Although some participants only provided one valid day of accelerometry data, the use of multi-level models using all available valid data from all participants provides unbiased estimates of daily MVPA and steps for each participant. These models adjusted for day of the week and day order of wear, both of which can affect steps and MVPA. Thus, we are not using the original data provided by the participant, but an adjusted average based on all available data from all participants. This method has been used successfully in other physical activity studies carried out by the authors and others (15). Durations of adequate wear-time to define habitual activity vary. Previous studies have often used a minimum of 3 to 5 days of accelerometry data (16), with as little as 480 minutes (17) per day to 600 minutes (18) per day of daily recording. As we outline in the paper, we chose to include all days of ≥540 minutes of registered accelerometry time during at least one day to lessen attrition bias and maximise inclusion of hard to reach groups, i.e., those from social housing who are likely to have lower compliance and record less PA data. This decision was part of the study design and was in our a priori Statistical Analysis Plan approved by the Steering Group (19). All physical activity related papers emanating from the ENABLE London study have followed this methodology (20-22). Moreover, as the reviewer correctly recognises, the sensitivity analysis referred to using 4 days of 9 hours in preference to 1 day, found no effect on the associations between PA and socio-demographic characteristics observed. In the current study, sensitivity analyses using 4+ days are not feasible as we analyse week days and weekend days separately, hence the need for inclusion of fewer days. We have added text to the Discussion (page 22, lines 481-488) to further justify the use of at least 1 day of 540 minutes.

8. Methods, Land-use mix and net residential density variables

Can the authors please provide further detail on the basis for the log-transformation of these variables? I haven’t seen this done before in other studies from other global settings. They state it is to “fit a comparable scale”. Comparable to what?

Response to reviewer

Walkability was derived by adding the z-scores of land use mix (LUM), residential density and street connectivity. LUM and residential density distributions were skewed. Because z-scores inform how far from the mean a data point is, it made sense to transform these 2 variables so that they better fit the normal distribution before deriving the z-scores. We have deleted the term “to fit a comparable scale” in the ‘Environmental variables’ section of the Methods (page 8, lines 161-162, and replaced this with “Land use mix and net residential density variables were log-transformed before deriving the z-scores due to their skewed distributions.”.

9. Methods

When explaining the rationale for the 1-KM network buffers, it might be good to state the average walking distance/time this represents (e.g., about a 10-15 minute walk from home in any direction for most people).

Response to reviewer

We have added that “a 1-km street network is equivalent to a 10-15-minute walk from home/centre of the buffer to its boundaries” to the ‘Environmental variables’ section of the Methods (page 8, lines 168-169).

10. Methods

Proximity to parks variable: other studies have found limited utility in these distance-based variables, including IPEN-adult, a 12 country study, where park density within 1-KM buffers was deemed a more useful predictor of physical activity. Did the authors consider using a buffer-based variable for parks too? Why was this discarded?

Response to reviewer

We agree with the reviewer that this is an important consideration. Because deriving built environment variables is time-consuming, we had to take a priori decisions as to which variables to create over others. The decision to use a distance-based rather than a density-based measure for park accessibility was based on the following.

First, a distance-based measure of park accessibility was likely to provide greater variability across our analysis sample than a density-based measure. In urban areas, where parks are a relatively sparse facility, a large portion of participants may not have had any parks falling within their 1km street-network home-centred buffer, leaving us with a considerably large group of participants with “no access”. The use of a distance-based measure overcomes this issue by attributing a quantitative value of accessibility even to those who do not have a park within the close vicinity of their home.

Second, London parks have a great deal of heterogeneity in size (from an average of 5 hectares for local parks to an average of 53 hectares for metropolitan parks) and in the type of facilities they provide. Therefore, there is a great deal of variation on functionality and utility. Size and facility types are known to influence how far individuals are ready to travel to use these facilities. Hence, a density-based measure of accessibility using a single buffer for the 3 types of parks would have not captured this attractivity dimension. A distance-based measure with models fitted separately for the 3 types of park appeared a better strategy than density-based measures for this setting.

11. Methods

I find it very interesting that car ownership was not deemed as a relevant confounder in this analysis, as this contradicts what we know about this variable in other parts of the world.

Response to reviewer

London has a highly developed and well-integrated public transport system with 64% of all trips undertaken by walking, cycling or public transit (TfL Travel in London Report 11, 2018 – http://content.tfl.gov.uk/travel-in-london-report-11.pdf). In addition, it has high street connectivity and high residential density. Hence, car use is much lower compared to elsewhere and we did not consider car ownership to be a major confounder. However, to respond constructively to the reviewer’s comment, we have carried out a sensitivity analysis to examine the effect of adjustment for car ownership on the main findings and can confirm that this has no material effect. We have added a comment to this effect (Statistical analyses, page 11, lines 242-244; Results, page 16, lines 353-354).

12. Statistical analysis

The first phrase is not clear. Authors mention a series of variables “were examined”. This could mean anything. Please be clear on the actual analytic procedures applied for “examining” these variables.

Response to reviewer

Thank you for highlighting this. On reflection, we feel this sentence is not needed and have removed it. We have also used this opportunity to re-write the Statistical Analysis section of the Methods (page 11) to make this clearer and more concise (see Response to comment #6 above).

13. Statistical analysis

Was a test for interaction actually performed to determine of day of the week (weekend vs. weekday) actually modifies the effect observed for some of these BE variables on the PA outcomes?

Response to reviewer

Literature suggests that patterns of PA are different on weekdays and weekends, and built environment variables may also have different effects on PA on weekdays compared with weekend days. Hence, average daily steps and MVPA were calculated separately for weekday and weekend days. We assessed if there was evidence of difference in associations on weekdays and weekend days by carrying out a Z-test to compare the coefficients on weekdays and weekend days. We have added the following to the Statistical Analysis section on page 11 (lines 227-229): 

“Differences in weekday and weekend physical activity associations with built environment characteristics from the separate models were formally assessed using Z-tests.”.

On page 14 (lines 315-322), we have added the following paragraph to the ‘Built environment and PA-level associations’ section in the Results, after consideration of ‘On weekdays - ’ and ‘On weekend days - ’ effects:

“Comparison of weekday versus weekend day effects – There was formal evidence of a difference between weekday and weekend day physical activity associations with the built environment in some of the sociodemographic adjusted models (Model 2). Specifically, (1) increased steps on a weekend day compared with weekday per unit increase in walkability (p=0.041), (2) increased steps and MVPA at weekends versus weekdays per km increase in distance to local park (p=0.015, p=0.026 respectively), and (3) increased steps on a weekday compared with a weekend day associated with lower public transport accessibility (p=0.02). All other differences between weekend versus weekday physical activity associations with the built environment were not formally statistically significant (p>0.05).”.

14. Statistical analysis

The use of multilevel models seems adequate, but I am surprised at such a high ICC (0.3, wow!). It would be important to report what percentage of the sample shared a household with another sample participant. Most studies of this type only accept one participant per household to avoid this level of clustering. 

Also, did the authors explore and, if needed, account for clustering at the neighborhood level (e.g., people residing in the same zip-code or small administrative geospatial unit?).

Response to reviewer

The participants invited to take part in the study were individuals and families who were looking to move to East Village. Thus, households with more than one participant were a feature of the study and we encouraged more than one member of a household to take part. At baseline, we recruited 1278 adults from 1006 households, with 40% sharing a household with another participant. Multi-level models were necessary to allow for clustering at household level, this approach was driven by the study design rather than the ICC, and was determined by our a priori statistical analysis plan, which was approved by our Steering Committee. The suggestion of clustering at the neighbourhood level is an interesting point but we did not explore this in our data, partly because controlling for neighbourhood effects by allowing for area clustering, might potentially dampen local influences of the built environment.

15. Statistical analysis

Could the authors explicitly state their hypotheses for the two different type of analyses proposed for examining the role of the BE in explaining PA-SES differences? I can follow the rational for the effect-measure modification analysis (testing for interaction terms of BE X SES variables). The simply adjustment by BE variables of the effect of SES on PA seems a repetition from the first modeling step, and implies confounding vs. effect measure modification, when, per the introduction, it seems that the authors have an effect modification hypothesis to begin with.

Response to reviewer

Following the reviewer’s suggestion, we have clarified in the last paragraph of the Introduction the aims of the study (pages 4-5, lines 88-95). Especially, we have better highlighted that the two pathways explored to examine the role of the BE in explaining PA-SES differences are (i) a moderating effect and (ii) a deprivation-amplification effect. The text now reads: 

“The aim of this London (UK) based study is twofold. First, to assess whether the residential built environment is associated with the number of daily steps, and the amount of daily MVPA (min) accumulated, on weekdays and weekend days. Second, to explore two pathways in which the built environment may contribute to household-level socio-economic differences in PA levels on weekdays and weekend days: (i) a moderating effect, whereby different socio-economic groups relate differently to the physical activity facilities available in their neighbourhood; (ii) a deprivation-amplification effect (2), by which limited access to PA-promoting facilities for disadvantaged individuals translate into lower levels of PA.”.

16. Statistical analysis

Also, the authors don’t mention which test for interaction and alpha level they used in their analysis (hopefully this was not just based on the p-value of the interaction term in the regression model – but was the result of a formal test for interaction – please clarify).

Response to reviewer

Post-estimation commands were used to look at the effects of adding interaction terms. We have edited the Statistical Analysis section on page 11 (lines 233-236), as follows:

“First, to examine the moderating effect of housing group on the association between the built environment and PA levels, interaction terms for housing group and built environment variables were added to the models and Stata post-estimation commands (testparm) were used to assess their statistical significance.”.

17. Statistical analysis

Outcome variables in models (steps and mins of MVPA/day): were these transformed in any way for achieving normality? Physical activity data are usually not normally distributed, hence the use of linear regression models usually requires a data transformation step. Alternatively, one could consider the use of logistic regression models (multinomial, politomous, or binary).

Response to reviewer

We have checked for normally distributed data before running our linear regression models. As we limited our analysis to those with days of ≥540 minutes of physical activity data, the distributions of average daily steps and average daily MVPA appeared normally distributed. We also checked the distribution of the residuals, which were also normally distributed. Hence, we feel confident that our methods of analysis are appropriate. We have added a sentence to the statistical methods section (lines 229-230) to confirm that the analytic approach was appropriate, as follows: “Residuals from all models were checked for assumption of normality to confirm the analytic approach as appropriate.”.

18. Statistical analysis

Did the authors examine potential multicollinearity in their models? This is usually a concern with some built environment variables in a single model. Please outline the methods via which this was examined.

Response to reviewer

Multicollinearity was tested in our models using the Variance Inflation Factor. VIF values did not exceed 2, downplaying concerns over multicollinearity. We have amended the Results section (lines 287-290) of the manuscript where concerns over multicollinearity had already been raised, as follows: “Although adjustment for other residential built environmental factors led to an inflation of the regression estimate and a widening of the confidence interval compared with models adjusting only for sociodemographics, Variance Inflation Factor (VIF) values did not exceed 2, downplaying concerns over multicollinearity.”.

19. Results, Table 2

Table 2 title: Should read: “Regression estimates for the association between residential built environment favors with physical activity outcomes (daily steps and mins/wk of MVPA), among… XXX [insert study name, year of data collection, etc.)”. It is customary to list the independent variable first and the dependent variable last, not the other way around.

Response to reviewer

Thank you for this suggestion. We have changed the title for Table 2 to ‘Regression estimates for the association between residential built environment variables with physical activity outcomes (daily steps and daily minutes of MVPA) in the ENABLE London study’ as suggested. Numbers and dates of data collection are now in the footnote.

20. Results

For all table/figure titles: add information on the study and sample, and year data were collected, as tables and figures should be “stand-alone” items. 

Response to reviewer

We have changed the titles of the tables to add “ENABLE London study” at the end of the titles and added footnotes to indicate numbers of participants and years of data collection.

21. Results

Figure 2 is very nice and helpful for the reader.

Response to reviewer

Thank you for this comment.

22. Results

I like the way the results for the examination of the role of the BE on the PA-SES relation are presented by two potential pathways. However, it would have been much more helpful to lay out the two hypothesized pathways since the introduction and methods (Statistical analysis) section, so the readers understand why both approaches (test for effect modification vs. BE as a confounder of the SES-PA relation) were explored. Otherwise the “two pathways” seem more a bit of an afterthought. For this purpose, it might be good for the authors to present a figure with the hypothetical DAGs of the two pathways they are exploring, a priori (i.e., before the results section).

Response to reviewer

We thank the reviewer for this suggestion. We have amended the Methods section (lines 232-239) so that the analyses described here relate more directly to the aims they serve. The paragraph now reads: “Further models examined two ways in which built environment variables may contribute to socio-economic differences in PA levels. First, in order to examine the moderating effect of housing group on the association between the built environment and PA levels, interaction terms for housing group and built environment variables were added to the models and Stata post-estimation commands (testparm) were used to assess their statistical significance. Second, in order to examine whether differences in access to PA facilities across housing groups translate into differences in levels of PA, attenuation in PA outcomes for each housing group was examined following adjustment for each built environment variable.”.

23. Discussion

I enjoyed reading the interpretation of the results regarding the counter-intuitive finding on longer distance to small parks being associated to higher levels of MVPA on weekends. This makes a lot of sense and it is not something the readers could have derived just by seeing the estimates. The discussion on the public transit results should be re-framed. I would suggest not opening with a statement on how they found a positive association with access to transit and PA, given that in reality once the models were adjusted this was not significant (the authors do state this, but I just don’t see the need to open with a statement of a positive relation). I think a deeper discussion is warranted to explain why their findings are inconsistent with most other reports of transit and PA. It is counterintuitive that those with lower availability to public transit have higher levels of PA, in a model that is also adjusted for SES. I wonder if adjusting for car ownership would have made a difference for this specific variable [access to transit] (?). The discussion on this topic should also include some reflection on the difference in measures of access to transit. Most other studies have relied on transit stop density measures (count of public transit stops within a buffer), and thus, are not truly comparable to the measure used in this study.

Response to reviewer

Following the reviewer’s suggestion, we have now revamped the discussion on accessibility to public transport (see lines 400-423). Especially, we have (i) better put our findings in light with previous literature findings, (ii) discussed whether accessibility to a motor vehicle may partly explain the association, and (iii) acknowledged that our variable is reflecting the concept of accessibility in a slightly different way to more simple density-based measures commonly used, as follows:

“Positive associations between accessibility to public transport and PA levels have been found worldwide (23), including in the UK (24), suggesting that interest in use of public transport may decrease with decreasing accessibility. For instance, Sallis et al. found that accessibility to public transport, measured as the density of public transport access points within a 1-km buffer centred on the home address, positively relates to adult PA levels in 14 cities worldwide, including Stoke-on-Trent, UK (24). Yet, our findings did not fully align with this conclusion, as the positive association between accessibility to public transport and steps and MVPA found on weekends failed to reach significance after adjustment for sociodemographic factors. More surprisingly, the association becomes negative on weekdays, with participants experiencing a low accessibility to public transport taking more daily steps compared to those with high accessibility. To our knowledge, such findings have not been reported in the literature and raise several questions. Participant’s heavy reliance on public transportation for inescapable weekday activities, like work, may explain why those with poor accessibility to public transport walk more on weekdays than those who live closer. Examining whether this negative association between public transport accessibility and PA on weekdays holds for individuals who have alternative ways of travelling (e.g., car) offers a further avenue for research. On a different note, our measure of public transport accessibility is not strictly comparable to density-based measures (simple count of public transit stops within a given area) used in other studies, in that it also accounts for frequency and reliability of service. Yet, our variable reflects a concept of public transport accessibility not that different to more basic density-based variables and is unlikely to be responsible for inverting the direction of the association.”.

24. Limitations paragraph

The use of a single day of accelerometry data with at least 9 hours is not in line with previous studies and should be acknowledged.

Response to reviewer

We have highlighted in the Strengths/Limitations section of the Discussion that our choice to include all days of ≥540 minutes of registered accelerometry time during at least one day was made to lessen attrition bias and maximise inclusion of hard to reach groups (i.e., those from social housing who are likely to have lower compliance and record less PA data). We have also presented this choice in light of previous studies that used accelerometer data. The other crucial point is that weekend estimates can be at best based on only two days of data. The text (lines 481-488) now reads: “Durations of adequate wear-time to define habitual activity vary. Previous studies have often used a minimum of 3 to 5 days of accelerometry data (16), with as little as 480 minutes (17) per day to 600 minutes (18) per day of daily recording. We chose to include all days of ≥540 minutes of registered accelerometry time during at least one day to lessen attrition bias and maximise inclusion of hard to reach groups, i.e., those from social housing who are likely to have lower compliance and record less PA data. This decision was part of the study design and was in our a priori Statistical Analysis Plan approved by the ENABLE London Steering Committee (19).”.

25. Limitations paragraph

The transit access measure, although apparently more sophisticated than what others use, is not comparable to what most researchers use when examining the role of transit infrastructure on PA.

Response to reviewer

We have raised this concern in the Discussion (lines 418-423), as follows:

“On a different note, our measure of public transport accessibility is not strictly comparable to density-based measures (simple count of public transport stops within a given area) used in other studies, in that it also accounts for frequency and reliability of service. Yet, our variable reflects a concept of public transport accessibility not that different to more basic density-based variables and is unlikely to be responsible for inverting the direction of the association.”.

26. Limitations paragraph

The Uncertain Geographic Context Problem and this study’s inability to address it should be listed as a limitation.

Response to reviewers

We have referred to the Uncertain Geographic Context Problem in the limitation paragraph (lines 496-504), as follows:

“Moreover, partly because we overlooked non-residential exposures to the built environment, we were not able to align with calls for addressing the Uncertain Geographic Context Problem (i.e., the extent to which areal units of measurement deviate from the geographic context truly experienced by individuals) (25). Although the spatial unit used to derive our environmental exposure complies with the wide majority of studies that have looked at the way the residential built environment relates to health behaviours (i.e. using an home-centred buffer within a walkable distance radius), focusing on the residential neighbourhood only may have led to misestimation of the association between the residential environment and health behaviours (26).”

27. Limitations paragraph

Another major limitation is the authors simply assumed differences in the relation of the BE and PA by weekday (effect modification by type of weekday), which is reasonable, but a stronger study analysis would have included a confirmation of this with a test for interaction, followed by the stratified analysis.

Response to reviewers

As highlighted in our response to comment #13, models were fitted separately for weekends and weekdays based on literature searches suggesting varying patterns of PA over the week. We assessed if there was evidence of difference in associations on weekend days and weekdays by carrying out a Z-test to compare the coefficients on weekend days and weekdays. This has been added to the ‘Statistical Analysis’ section of the Methods (page 11, lines 227-229).

28. Limitations paragraph

Finally, selection bias due to a) the response rate, and b) the eligibility criteria (this is not a representative sample of London residents, rather, it targets people with a desire to move) needs to be fully disclosed, with a description of how these factors are likely limiting the external validity of the study. In particular, because a reason for moving might be the BE features of the new neighborhood where people are hoping to move to.

Response to reviewers

As highlighted in our response to comment #5, we have acknowledged this in the Limitation paragraph (lines 488-494), as follows:

“This study draws on the baseline sample of a natural experiment, where recruits were deemed eligible if they were seeking to relocate to East Village. Such sampling strategy limits external validity, as the participants may not be truly representative of the London population as a whole. Moreover, the cross-sectional study design is prone to selection bias (i.e., those living in less walkable neighbourhoods are intrinsically different to those who do not), and therefore restricts interpretations about direction of effects.”.

Supporting references

1. Ding J, Wai KL, McGeechan K, Ikram MK, Kawasaki R, Xie J, et al. Retinal vascular caliber and the development of hypertension: a meta-analysis of individual participant data. J Hypertens. 2014;32(2):207-15.

2. Macintyre S, Ellaway A, Cummins S. Place effects on health: how can we conceptualise, operationalise and measure them? Soc Sci Med. 2002;55(1):125-39.

3. Frohlich KL, Corin E, Potvin L. A theoretical proposal for the relationship between context and disease. Sociology of health & illness. 2001;23(6):776-97.

4. Ogilvie D, Lamb KE, Ferguson NS, Ellaway A. Recreational physical activity facilities within walking and cycling distance: Sociospatial patterning of access in Scotland. Health & Place. 2011;17(5):1015-22.

5. Higgs G, Langford M, Norman P. Accessibility to sport facilities in Wales: A GIS-based analysis of socio-economic variations in provision. Geoforum. 2015;62:105-20.

6. Pliakas T, Wilkinson P, Tonne C. Contribution of the physical environment to socioeconomic gradients in walking in the Whitehall II study. Health Place. 2014;27:186-93.

7. Barbosa O, Tratalos JA, Armsworth PR, Davies RG, Fuller RA, Johnson P, et al. Who benefits from access to green space? A case study from Sheffield, UK. Landscape and Urban Planning. 2007;83(2-3):187-95.

8. Hillsdon M, Panter J, Foster C, Jones A. Equitable access to exercise facilities. Am J Prev Med. 2007;32(6):506-8.

9. Forsyth A, Oakes JM, Lee B, Schmitz KH. The built environment, walking, and physical activity: Is the environment more important to some people than others? Transportation research part D: transport and environment. 2009;14(1):42-9.

10. Manaugh K, El-Geneidy A. Validating walkability indices: How do different households respond to the walkability of their neighborhood? Transportation research part D: transport and environment. 2011;16(4):309-15.

11. Van Dyck D, Cerin E, De Bourdeaudhuij I, Salvo D, Christiansen LB, Macfarlane D, et al. Moderating effects of age, gender and education on the associations of perceived neighborhood environment attributes with accelerometer-based physical activity: The IPEN adult study. Health Place. 2015;36:65-73.

12. Winters M, Barnes R, Venners S, Ste-Marie N, McKay H, Sims-Gould J, et al. Older adults' outdoor walking and the built environment: does income matter? BMC Public Health. 2015;15:876.

13. Cummins S, Curtis S, Diez-Roux AV, Macintyre S. Understanding and representing 'place' in health research: a relational approach. Soc Sci Med. 2007;65(9):1825-38.

14. Freedson PS, Melanson E, Sirard J. Calibration of the Computer Science and Applications, Inc. accelerometer. Med Sci Sports Exerc. 1998;30(5):777-81.

15. Harris T, Kerry SM, Limb ES, Victor CR, Iliffe S, Ussher M, et al. Effect of a Primary Care Walking Intervention with and without Nurse Support on Physical Activity Levels in 45- to 75-Year-Olds: The Pedometer And Consultation Evaluation (PACE-UP) Cluster Randomised Clinical Trial. PLoS Med. 2017;14(1):e1002210.

16. Trost SG, McIver KL, Pate RR. Conducting accelerometer-based activity assessments in field-based research. Med Sci Sports Exerc. 2005;37(11 Suppl):S531-43.

17. Miller GD, Jakicic JM, Rejeski WJ, Whit-Glover MC, Lang W, Walkup MP, et al. Effect of varying accelerometry criteria on physical activity: the look ahead study. Obesity (Silver Spring). 2013;21(1):32-44.

18. Katzmarzyk PT, Champagne CM, Tudor-Locke C, Broyles ST, Harsha D, Kennedy BM, et al. A short-term physical activity randomized trial in the Lower Mississippi Delta. PLoS One. 2011;6(10):e26667.

19. Ram B, Nightingale CM, Hudda MT, Kapetanakis VV, Ellaway A, Cooper AR, et al. Cohort profile: Examining Neighbourhood Activities in Built Living Environments in London: the ENABLE London-Olympic Park cohort. BMJ Open. 2016;6(10):e012643.

20. Limb ES, Procter DS, Cooper AR, Page AS, Nightingale CM, Ram B, et al. The effect of moving to East Village, the former London 2012 Olympic and Paralympic Games Athletes' Village, on mode of travel (ENABLE London study, a natural experiment). Int J Behav Nutr Phys Act. 2020;17(1):15.

21. Nightingale CM, Limb ES, Ram B, Shankar A, Clary C, Lewis D, et al. The effect of moving to East Village, the former London 2012 Olympic and Paralympic Games Athletes' Village, on physical activity and adiposity (ENABLE London): a cohort study. Lancet Public Health. 2019;4(8):e421-e30.

22. Nightingale CM, Rudnicka AR, Ram B, Shankar A, Limb ES, Procter D, et al. Housing, neighbourhood and sociodemographic associations with adult levels of physical activity and adiposity: baseline findings from the ENABLE London study. BMJ Open. 2018;8(8):e021257.

23. Li F, Harmer PA, Cardinal BJ, Bosworth M, Acock A, Johnson-Shelton D, et al. Built environment, adiposity, and physical activity in adults aged 50-75. Am J Prev Med. 2008;35(1):38-46.

24. Sallis JF, Cerin E, Conway TL, Adams MA, Frank LD, Pratt M, et al. Physical activity in relation to urban environments in 14 cities worldwide: a cross-sectional study. The Lancet. 2016;387(10034):2207-17.

25. Kwan M-P. The uncertain geographic context problem. Annals of the Association of American Geographers. 2012;102(5):958-68.

26. Chaix B, Duncan D, Vallee J, Vernez-Moudon A, Benmarhnia T, Kestens Y. The "residential" effect fallacy in neighborhood and health studies: formal definition, empirical identification, and correction. Epidemiology. 2017.

---

## [Decision Letter · Decision Letter 1]

27 Jul 2020

Weekend and weekday associations between the residential built environment and physical activity: findings from the ENABLE-London Study

PONE-D-19-33687R1

Dear Dr. Owen,

We’re pleased to inform you that your manuscript has been judged scientifically suitable for publication and will be formally accepted for publication once it meets all outstanding technical requirements.

Kind regards,

Adewale L. Oyeyemi, Ph.D

Academic Editor

PLOS ONE

Additional Editor Comments (optional):

Reviewers' comments:

Reviewer's Responses to Questions

**Comments to the Author**

1. If the authors have adequately addressed your comments raised in a previous round of review and you feel that this manuscript is now acceptable for publication, you may indicate that here to bypass the “Comments to the Author” section, enter your conflict of interest statement in the “Confidential to Editor” section, and submit your "Accept" recommendation.

Reviewer #1: All comments have been addressed

2. Is the manuscript technically sound, and do the data support the conclusions?

Reviewer #1: Yes

3. Has the statistical analysis been performed appropriately and rigorously? 

Reviewer #1: Yes

4. Have the authors made all data underlying the findings in their manuscript fully available?

Reviewer #1: No

5. Is the manuscript presented in an intelligible fashion and written in standard English?

Reviewer #1: Yes

6. Review Comments to the Author

Reviewer #1: The authors addressed all of my comments in a thoughtful manner. I am satisfied with this new version of the manuscript. Thank you again for the opportunity to review this important paper.

7. PLOS authors have the option to publish the peer review history of their article (what does this mean?). If published, this will include your full peer review and any attached files.

Reviewer #1: **Yes: **Deborah Salvo

---

## [Editor Report · Acceptance letter]

3 Aug 2020

PONE-D-19-33687R1 

Weekend and weekday associations between the residential built environment and physical activity: findings from the ENABLE-London Study 

Dear Dr. Owen:

I'm pleased to inform you that your manuscript has been deemed suitable for publication in PLOS ONE. Congratulations! Your manuscript is now with our production department. 

Kind regards, 

on behalf of

Dr. Adewale L. Oyeyemi 

Academic Editor

PLOS ONE